# SCF$^{Cdc4}$ ubiquitin ligase regulates synaptonemal complex formation during meiosis

Zhihui Zhu, Mohammad Bani Ismail, Miki Shinohara, Akira Shinohara

Homologous chromosomes pair with each other during meiosis, culminating in the formation of the synaptonemal complex (SC), which is coupled with meiotic recombination. In this study, we showed that a meiosis-specific depletion mutant of a cullin (Cdc53) in the SCF (Skp-Cullin-F-box) ubiquitin ligase, which plays a critical role in cell cycle regulation during mitosis, is deficient in SC formation. However, the mutant is proficient in forming crossovers, indicating the uncoupling of meiotic recombination with SC formation in the mutant. Furthermore, the deletion of the *PCH2* gene encoding a meiosis-specific AAA+ ATPase suppresses SC-assembly defects induced by *CDC53* depletion. On the other hand, the *pch2 cdc53* double mutant is defective in meiotic crossover formation, suggesting the assembly of SC with unrepaired DNA double-strand breaks. A temperature-sensitive mutant of *CDC4*, which encodes an F-box protein of SCF, shows meiotic defects similar to those of the *CDC53*-depletion mutant. These results suggest that SCF$^{Cdc4}$, probably SCF$^{Cdc4}$-dependent protein ubiquitylation, regulates and collaborates with Pch2 in SC assembly and meiotic recombination.

## Introduction

Meiosis is a specialized form of cell division that generates haploid gametes (Petronczki et al, 2003; Marston & Amon, 2004). Upon entry into meiosis, cells undergo DNA replication followed by two rounds of nuclear division. During meiosis I, homologous chromosomes segregate to opposite poles. Crossovers (COs), which mean reciprocal exchanges between homologous chromosomes, are essential for the segregation of chromosomes during meiosis I by providing physical linkages between the chromosomes (Hunter, 2015; Gray & Cohen, 2016).

Meiotic prophase I exhibits drastic chromosome dynamics and morphogenesis. Homologous loci on two parental chromosomes pair with each other during prophase I (Zickler & Kleckner, 1999; Kleckner, 2006). Pairing culminates as the synapsis of homologous chromosomes, manifested by the formation of the synaptonemal complex (SC), which is a meiosis-specific chromosome structure (Zickler & Kleckner, 1999; Kleckner, 2006). SC contains the central region with polymerized transverse filaments, which are flanked by two homologous chromosomal axes with multiple chromatin loops referred to as lateral elements (LEs) (Cahoon & Hawley, 2016; Gao & Colaiacovo, 2018). SC assembly and disassembly are tightly regulated along the progression of meiosis. In the leptotene stage, a pair of sister chromatids is folded into a chromosome axis with chromatin loops, called axial elements (AEs). Leptonema is followed by zygonema, in which a short patch of SC is formed between homologous AEs. SC elongation occurs along chromosomes, resulting in the formation of full-length SCs at pachynema, where AEs are referred to as LEs. In some organisms, AE elongation is coupled with SC formation, whereas in others, LE/AE formation proceeds via SC formation. SCs are then disassembled in diplonema before the onset of anaphase-I. Importantly, SC formation is tightly coupled with meiotic recombination in most organisms, including budding yeast and mammals. Mutants defective in meiotic recombination show defects in SC formation, for example, the *spo11* and *dmc1* mutants (Giroux et al, 1989; Bishop et al, 1992; Baudat et al, 2000; Romanienko & Camerini-Otero, 2000), which are deficient in the formation of DNA double-strand breaks (DSBs) and strand exchange between homologous DNAs, respectively. On the other hand, in fruit flies and nematodes, SC formation is independent of the initiation of meiotic recombination (Dernburg et al, 1998; McKim & Hayashi-Hagihara, 1998).

Synapsis of homologous chromosomes, leading to SC formation, initiates at a specific site along chromosomes, which likely corresponds to the site of meiotic recombination. In budding yeast, the evolutionarily conserved Zip, Msh, Mer (ZMM)/synapsis initiation complex proteins, including Zip1, Zip2, Zip3, Msh4, Msh5, Mer3, Spo16, Spo22/Zip4, and Pph3, can promote SC assembly as well as CO formation (Hollingsworth et al, 1995; Chua & Roeder, 1998; Nakagawa & Ogawa, 1999; Agarwal & Roeder, 2000; Borner et al, 2004; Hochwagen et al, 2005; Tsubouchi et al, 2006; Shinohara et al,

Institute for Protein Research, Osaka University, Suita, Osaka, Japan

Correspondence: ashino@protein.osaka-u.ac.jp
Zhihui Zhu's present address is College of Plant Science & Technology, Huazhong Agricultural University
Mohammad Bani Ismail's present address is College of Science, King Faisal University
Miki Shinohara's present address is Graduate School of Agriculture, Kindai University

2008). ZMM proteins localize to chromosomes as a large protein ensemble, which is detected by immunostaining for SC assembly through the deposition of Zip1, a yeast transverse filament protein, into arrays in the central region of the SC (Sym et al, 1993; Sym & Roeder, 1995). Zip1 polymerization is promoted by the action of a complex containing Ecm11 and Gmc2 as a component of the SC central region (Humphryes et al, 2013; Voelkel-Meiman et al, 2013). AEs/LEs contain several meiosis-specific proteins, including Red1, Hop1, and Mek1/Mre4 kinase (Rockmill & Roeder, 1988, 1990; Hollingsworth et al, 1990; Leem & Ogawa, 1992) as well as a cohesin complex containing a meiosis-specific kleisin Rec8 (Klein et al, 1999). Rec8, Hop1, and Red1 are axis components evolutionarily conserved among species and are found as REC8, HORMAD1/2, and SYCP2/3 in mammals, respectively (Eijpe et al, 2003; Wojtasz et al, 2009; West et al, 2019). How AEs or meiotic chromosome axes, which may be independent of SC elongation, are assembled remains largely unknown.

Protein modifications mediated by small proteins, such as ubiquitin and small ubiquitin-like modifier protein (SUMO), regulate various biological processes during mitosis and meiosis. SUMOlyation is involved in SC formation (Nottke et al, 2017). SUMO localizes in the SC, both the SC central region and the axes, in budding yeast (Cheng et al, 2006; Hooker & Roeder, 2006; Voelkel-Meiman et al, 2013), and both SUMO and ubiquitin are present on the axes and SC central region in mouse spermatocytes (Rao et al, 2017). Budding yeast Ecm11 present in the central region of SCs is SUMOlyated (Humphryes et al, 2013; Voelkel-Meiman et al, 2013), and the amounts of SUMOlyated Ecm11 correlate with SC elongation (Leung et al, 2015). In mice, a SUMO ligase, Rnf212, and a ubiquitin ligase, Hei10, antagonize each other for meiotic recombination (Qiao et al, 2014). Moreover, the proteasome is localized on SCs in budding yeast, nematodes, and mice (Ahuja et al, 2017; Rao et al, 2017), suggesting a role for ubiquitin-dependent proteolysis in meiotic chromosome metabolism.

Two major ubiquitin ligases, the Skp-Cullin-F-box (SCF) and anaphase promoting complex/cyclosome (APC/C), play an essential role in the mitotic cell cycle (Feldman et al, 1997; Skowyra et al, 1997; Yu et al, 1998; Zachariae et al, 1998). In budding yeast meiosis, APC/C with either Cdc20 or Cdh1 promotes the timely transition of metaphase/anaphase I and II (Pesin & Orr-Weaver, 2008; Cooper & Strich, 2011). A meiosis-specific APC/C activator, Ama1, regulates the duration of prophase I (Okaz et al, 2012), which is negatively controlled by the APC/C subunit, Mnd2 (Oelschlaegel et al, 2005; Penkner et al, 2005).

In budding yeast, a core SCF, which is composed of Rbx1/Hrt1 (RING finger protein), Cdc53 (cullin), and Skp1, binds various F-box adaptor proteins, including Cdc4, Grr1, and Met30 (Willems et al, 2004; Nakatsukasa et al, 2015). These F-box proteins determine the substrate-specificity of the complex. SCF with Cdc4, referred to as SCF$^{Cdc4}$, mediates the ubiquitylation of G1 cyclin(s) and a Cdk inhibitor, Sic1, at the G1/S transition (Koivomagi et al, 2011). SCF also ubiquitylates Cdc6, which is essential for the initiation of DNA replication (Perkins et al, 2001). On the other hand, little is known about the role of the SCF during prophase I. A previous report indicated the role of SCF$^{Cdc4}$ in premeiotic DNA replication through Sic1 degradation (Sedgwick et al, 2006).

In this study, we analyzed the roles of the SCF ubiquitin ligase in yeast meiosis by characterizing a meiosis-specific depletion mutant of Cdc53 and found that Cdc53 is indispensable for SC formation and progression into anaphase I. Moreover, a mutant of the *PCH2* gene, which encodes a meiosis-specific AAA+ ATPase (San-Segundo & Roeder, 1999; Borner et al, 2008; Chen et al, 2014), suppresses SC-assembly defects by Cdc53 depletion. A temperature-sensitive *cdc4* mutant also showed meiotic defects similar to those of *CDC53* depletion. We propose that SCF$^{Cdc4}$ regulates proper SC assembly by counteracting the Pch2-dependent negative control on SC assembly.

# Results

### Depletion of Cdc53 induces meiosis I arrest

The SCF complex plays a critical role in the cell cycle control of mitosis (Willems et al, 2004). However, its role in meiosis is largely unknown because genes encoding core components of the complex (e.g., Rbx1, Cdc53, and Skp1) (Willems et al, 1999, 2004) are essential for vegetative growth of budding yeast, *Saccharomyces cerevisiae*. To determine the role of the SCF complex in budding yeast, we constructed a strain that depletes an SCF component specifically during meiosis by replacing the target gene promoter with the *CLB2* promoter whose activity is down-regulated during meiosis (Lee & Amon, 2003). In a strain with *pCLB2-HA-CDC53*, hereafter, *CDC53mn* (meiotic null), we could efficiently reduce the cullin component of the SCF, Cdc53 (Fig 1A), whose steady-state level did not change significantly during meiosis (Fig S1A). In the *CDC53mn* mutant, the level of Cdc53 began to decrease at 2 h after induction of meiosis, and a small amount of Cdc53 protein was detectable after 4 h. We checked the amounts of two known SCF$^{Cdc4}$ substrates, Sic1 and Cdc6. In wild-type cells, the level of Sic1 decreases at 0–2 h of incubation with sporulation medium (SPM) (Fig S1D), as shown previously (Sedgwick et al, 2006). In the *CDC53mn* mutant, Sic1 was still present at 4 h of meiosis but disappeared at 6 h, suggesting a delay in its degradation. On the other hand, Cdc6, which is very unstable after 4 h in wild-type cells (Perkins et al, 2001), was present at late time points, such as 12 h, in the mutant (Fig S1D). These results indicate that in the *CDC53mn* mutant, SCF activity is retained in the very early phase of meiotic prophase I, which is sufficient for triggering the premeiotic S phase but is largely decreased during further incubation.

Although the *CDC53mn* mutant showed normal growth during mitosis, it showed various defects during meiosis. The *CDC53mn*-mutant cells showed a delayed onset of meiotic DNA replication by ~2 h compared with the wild-type cells (Fig S1B). The delayed Sic1 degradation (Fig S1D) might explain the delay in the onset of the S phase in the mutant. DAPI staining showed that Cdc53-depleted cells arrested before meiosis I (Fig 1B). Aberrant recombination intermediates are known to trigger an arrest at the mid-pachytene stage, for example, the *dmc1* mutant (Bishop et al, 1992). However, the arrest in the *CDC53mn* mutant is independent of the recombination because the introduction of the *spo11-Y135F* mutation, which abolishes the formation of meiotic DSBs (Bergerat et al, 1997),

did not suppress the arrest induced by *CDC53* depletion (Fig S1C). We also checked the expression of Cdc5 (Polo-like kinase), which is induced after the mid-pachytene stage and decreases after meiosis I (Chu & Herskowitz, 1998; Clyne et al, 2003). *CDC53mn* cells expressed Cdc5 from 8 h, 2 h later than the wild type and maintained its expression at late time points, such as 12 h (Fig S1D), showing that the mutant exits the pachytene stage. High levels of Cdc5, whose degradation triggers the exit of meiosis I, in the mutant support that it does not undergo meiosis I. Tubulin staining revealed that the mutant showed delayed entry into metaphase-I with short spindles (Fig 1C). Even after 12 h, only half of the *CDC53mn* mutant cells contained short metaphase-I or anaphase-I spindles (Fig 1D), indicating an arrest at metaphase/anaphase I transition. These results indicate that Cdc53, probably the SCF, plays a pivotal role in the transition of metaphase I to anaphase I, suggesting the presence of a novel regulatory mechanism involved in the transition. The meiotic arrest induced by *CDC53* depletion is similar to that seen in meiosis-specific depletion of *CDC20*, which encodes an activator of APC/C for the onset of anaphase I (Lee & Amon, 2003). SCF may regulate APC/C activity during meiosis I, as seen in *Xenopus* oocytes (Nishiyama et al, 2007).

### *CDC53* depletion results in defective SC assembly

We examined the effect of Cdc53 depletion on meiotic prophase I events, such as SC formation by immunostaining of chromosome spreads. Zip1, a component in the central region of SCs, is widely used as a marker for SC formation (Sym et al, 1993). As a marker for meiotic DSB repair, we co-stained Rad51, a RecA-like recombinase (Shinohara et al, 1992; Bishop, 1994) with Zip1. In the wild type, Zip1 first appeared as several foci on chromosomes in early meiosis, such as the leptotene stage (Fig 1E). Then, short lines and later long lines of Zip1 were observed, which corresponded to zygotene (Fig 1E) and pachytene stages (Fig 1E), respectively. The *CDC53mn* mutant displayed defective SC assembly (Fig 1E and F). At early time points (2–4 h) in the mutant, dotty staining of Zip1 appeared without any delay compared with the wild type. This indicates a normal association of Zip1 with chromosomes at early meiosis I. The formation of short lines of Zip1 in the mutant was delayed by ~2–3 h, compared with that in the wild type (Fig 1E and F). Even at late time points (6 h), the mutant showed a significant reduction in the formation of fully elongated Zip1-lines (Fig 1F), indicating a defect in Zip1 elongation. Indeed, the mutant transiently accumulated an aggregate of Zip1, called poly-complex (PC), which is indicative of abnormal SC assembly (Sym et al, 1993) (Fig 1E and G). Although the mutant transiently accumulated these zygonema-like nuclei with short lines and PCs of Zip1, these Zip1 structures were almost dismantled at late time points, such as 10–12 h, in the mutant (Fig 1F and G). The disappearance of Zip1-positive cells was delayed by ~4 h in the mutant strain relative to the wild type, delaying prophase I by 2 h in the *CDC53mn* mutant compared with the wild type.

### *CDC53* depletion shows little defect in meiotic recombination

During meiotic prophase I, SC formation is tightly coupled with meiotic recombination (Alani et al, 1990; Padmore et al, 1991; Bishop

et al, 1992). Rad51 staining showed that the appearance of Rad51 foci on the chromosomes was delayed in the *CDC53mn*-mutant cells relative to wild-type cells (Fig 1E and H), probably because of delayed entry into meiosis (Fig S1B). However, the kinetics of Rad51-focus staining were similar to those in the wild type, with a delayed peak at 6 h (Fig 1H), suggesting a weak defect in DSB repair in the mutant.

To analyze meiotic recombination in the *CDC53*-depletion mutant, we analyzed the repair of meiotic DSBs and the formation of crossovers (COs) at a well-characterized recombination hotspot, the *HIS4-LEU2* (Cao et al, 1990) by Southern blotting (Fig 2A). In the wild type, DSBs appeared at ~3 h, peaked at 4 h, and then disappeared after 5 h (Fig 2B and C). Consistent with the delay in the onset of the meiotic S phase, the *CDC53mn* mutant showed a delay in DSB appearance by ~2 h relative to the wild type. When the delay in the S-phase entry was compensated, the kinetics of the appearance of meiotic DSBs in the mutant was similar to that in the wild type. On the other hand, there was a substantial delay (~1 h) in the disappearance of DSBs in the mutant, indicating a weak defect in meiotic DSB repair. These results suggest that the *CDC53mn* mutant was almost proficient in repairing meiotic DSBs. Indeed, the mutant was proficient in CO formation at this locus. In the wild type, COs started to form at 5 h and reached a maximum level of ~7.5% at 8 h (Fig 2D and E). Although the formation of COs in the mutant was delayed by ~3 h compared with the wild type, the final levels of COs in the mutant were indistinguishable from those in the wild type (Fig 2E). To confirm normal CO formation at the other locus in the *CDC53mn* mutant, we also analyzed the ectopic CO formation at a recombination hotspot of *URA3-ARG4* (Allers & Lichten, 2001). At this locus, ectopic COs were formed between the cassettes at the *leu2* and *his4* loci (Fig 2F). The *CDC53mn* mutant showed an ~2-h delay in the formation of the COs at the locus relative to the wild type (Fig 2G and H). The final level of CO products in the mutant was similar to that in the wild-type control (Fig 2H). In addition, we checked genome-wide DSB repair by examining chromosome bands using pulse-field gel electrophoresis (PFGE) (Fig S1E). The *CDC53mn* cells showed smeared chromosomal bands at 4 h, which were caused by DSB formation, and, as wild-type cells, recovered full chromosomal bands at 5 h. This result, together with Rad51 foci kinetics (Fig 1H), suggests that most of the DSBs were repaired under *CDC53* depletion conditions. Furthermore, these results indicate that SC formation was uncoupled with meiotic recombination in the *CDC53mn* mutant. Likely, full-length SCs are not required for the completion of CO formation in the absence of Cdc53.

### *CDC53* depletion results in altered assembly of some ZMM proteins

We examined the localization of ZMM proteins (Zip3, Zip2, Mer3, Spo22/Zip4, Msh4, and Msh5) that promote SC assembly and CO formation. As reported previously (Chua & Roeder, 1998; Agarwal & Roeder, 2000; Tsubouchi et al, 2006; Shinohara et al, 2008), these ZMM proteins show punctate staining during leptotene and pachytene stages (Figs 2I and S2A). We found that the staining of Zip3, Zip2, Mer3, and Spo22/Zip4 were altered in the *CDC53mn* mutant (Figs 2I, top and S2A). With a reduction of focus staining, the mutant accumulated PCs of Zip3, Zip2, Mer3, and Spo22/Zip4, which

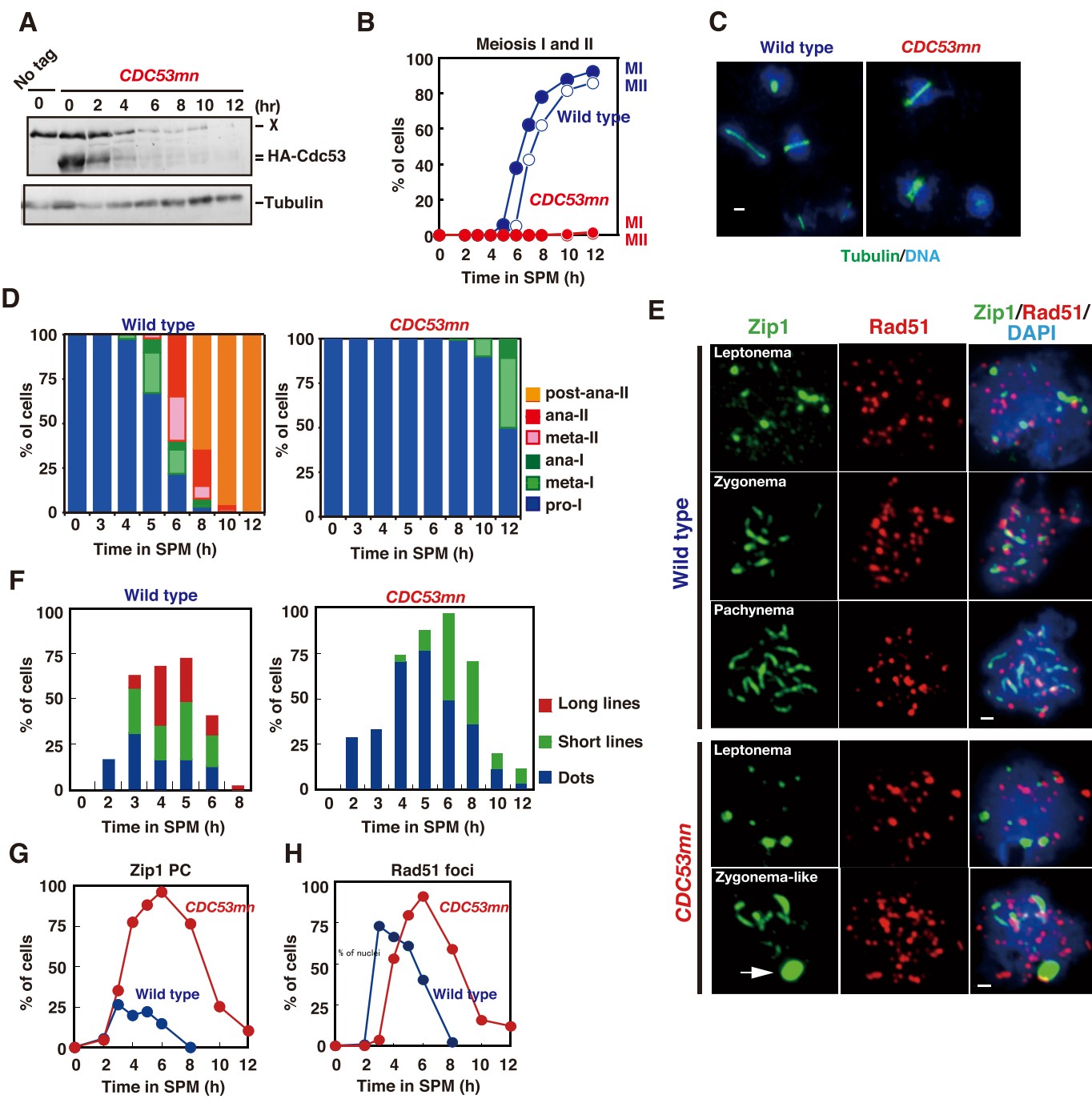

**Figure 1. Cdc53 depletion induces abnormal synaptonemal complex (SC).**
**(A)** Expression of Cdc53. Lysates obtained from wild-type (NKY1551, only at 0 h) and *CDC53mn* (ZHY94) cells at various time points during meiosis were analyzed by Western blotting using anti-HA (HA-Cdc53, upper) or anti-tubulin (lower) antibodies. "X" indicates a non-specific band reacted with anti-HA. **(B)** Meiotic cell cycle progression. The entry into meiosis I and II in wild-type and *CDC53mn* cells were analyzed by DAPI staining. The number of DAPI bodies in a cell was counted. A cell with 2, 3, and 4, and 3 and 4 DAPI bodies was defined as a cell that passed through meiosis I and meiosis II, respectively. The graph shows the percentages of cells that completed MI or MII at the indicated time points. More than 200 cells were counted at each time point. The representative results (*n* = 3) are shown; see the same results shown in Figs 3A and 6A. Wild type, MI, blue closed circles; wild type, MII, blue open circles; *CDC53mn* MI, red closed circles; *CDC53mn* MII, red open circles. **(C)** Tubulin staining in the *CDC53mn* mutant. Whole wild-type (5 h) and *CDC53mn* cells (8 h) were fixed and stained with anti-tubulin (green) and DAPI (blue). Representative images are shown. Bar = 2 μm. **(D)** Classification of tubulin/DAPI staining at each time point of meiosis in wild-type (left) and *CDC53mn* mutant (right) cells. Dot, short line, and long line tubulin-staining with single DAPI mass were defined as prophase I, metaphase I, and anaphase I, respectively, and are shown in different colors. Short and long tubulin staining were defined as metaphase II and anaphase II, respectively. At each time point, more than 100 cells were counted. The representative results are shown (*n* = 2). **(E)** Zip1 and Rad51 staining. Nuclear spreads from wild-type and *CDC53mn* mutant cells were stained with anti-Zip1 (green), anti-Rad51 (red), and DAPI (blue), and then categorized into different cell cycle stages. SCs of wild-type cells are shown in leptotene (Class I), zygotene (Class II), and pachytene (Class III) stages. Class II and II contained less than 10 and more than 10 Zip1 lines, respectively. SCs of *CDC53mn* mutants are shown in leptotene and zygotene-like stages (E). Representative images are shown. Zygotene-like *CDC53mn* cells

almost co-localized with Zip1 PCs (Figs 2I and S2A). These results indicate altered assembly of some ZMM proteins in the *CDC53mn* mutant.

Unlike the foci of Zip2, Zip3, Mer3, and Spo22/Zip4 proteins, the foci of Msh4 and Msh5, which form a hetero-dimeric MutSγ complex (Hollingsworth et al, 1995; Novak et al, 2001), appeared normal in the *CDC53mn* mutant compared with those in the wild type (Figs 2I and S2A). In the mutant, Msh4 and Msh5 foci appeared with a 2-h delay and disappeared with a 4-h delay relative to the wild type (Fig 2J). As in the wild type, the kinetics of Msh4 and Msh5 were similar to those of Rad51 in the mutant (Figs 1H and 2J). Notably, the *CDC53mn* mutant did not form any PCs containing Msh4 or Msh5 (Figs 2I and S2A), unlike other SC-defective mutants such as *zmm* (Shinohara et al, 2008, 2015). We also counted a steady-state number of Msh5 foci in the *CDC53mn* mutant. The average number of bright Msh5 foci at 6 h in the mutant (Fig 2K) was 30.7 ± 7.3 (n = 20), which was indistinguishable from that in the wild type (29.1 ± 6.1; Mann–Whitney *U* test, *P* = 0. 34) at 4 h. The normal assembly and disassembly of Msh4 and Msh5 could explain the proficiency of meiotic CO formation in the mutant (Fig 2E and H). This result supports the idea that, among ZMMs, Msh4-Msh5 is a key protein involved in CO formation, whose functions are distinct from other ZMMs (Shinohara et al, 2008; Pyatnitskaya et al, 2019). Moreover, Cdc53 plays a role in the proper assembly of the subsets of ZMM proteins other than Msh4 and Msh5 for SC assembly.

### Cdc53 and Zip3 distinctly work in SC formation and recombination

To determine the relationship between Cdc53 (ubiquitin ligase) and Zip3 (SUMO ligase), both of which are involved in SC assembly, we characterized the *CDC53mn zip3* double mutant. Like the *CDC53mn* single mutant, the *CDC53mn zip3* double mutant showed meiotic arrest (Fig 3A), which was different from the *zip3* mutant with delayed progression of the meiotic prophase (Agarwal & Roeder, 2000; Borner et al, 2004). Whereas both *CDC53mn* and *zip3* single mutants can repair meiotic DSBs with a substantial delay, *CDC53mn zip3* double mutant accumulated unrepaired DSBs at the *HIS4-LEU2* hotspot with hyper-resection at late time points (Fig 3B and C). The inability of the double mutant to repair the DSBs was confirmed by the accumulation of Rad51 foci at late time points, such as 12 h (Fig 3D and G), whereas the disappearance of Rad51 foci was seen in both *CDC53mn* and *zip3* single mutants. Moreover, the *CDC53mn zip3* double mutant formed lower levels of COs compared with the *zip3* single mutant (Fig 3E and F). Unlike the *CDC53mn* single mutant, the double mutant did not express Cdc5 as a marker for pachytene exit (Chu & Herskowitz, 1998) (Fig S2B), indicating an arrest at the mid-pachytene stage induced by the recombination checkpoint response to unrepaired DSBs. These results suggest that Cdc53 plays

a role in the efficient repair of meiotic DSBs, and thus CO formation, in the absence of Zip3.

SC formation in the *CDC53mn zip3* double mutant was analyzed by Zip1 staining. On the spreads of both the *CDC53mn* and *zip3* single mutants, short lines of Zip1 were often formed (Fig 3G and H). On the other hand, the *CDC53mn zip3* double mutant showed a reduced formation of short Zip1 lines compared with either single mutant (Fig 3H). In contrast to the single mutants, which showed disassembly of abnormal SCs at late time points, the double mutant did not show SC disassembly (Fig 3H), which was probably due to mid-pachytene arrest (Fig S2B). These results suggest that Cdc53 and Zip3 distinctly work for SC assembly and CO formation.

### *CDC53mn* mutant forms altered chromosome axis

Defective SC assembly in *CDC53mn* cells is due to either SC assembly per se or rapid turnover (precocious disassembly) of fully elongated SCs in the mutant. To distinguish these possibilities, we introduced an *ndt80* mutation, which blocks the disassembly of the SC by inducing mid-pachytene arrest (Fig S3A) (Xu et al, 1995). As shown previously (Xu et al, 1995), the *ndt80* single mutant accumulated full-length SCs (Fig S3B–D). The *ndt80* mutation weakly suppressed the SC-elongation defects in the *CDC53mn* mutant only at late time points, such as 10 h (Fig S3D). About 20% of the *CDC53mn ndt80* double mutants showed long Zip1 lines at late time points, whereas a few long Zip1 lines were transiently formed upon Cdc53 depletion, indicating that the SC assembly defect in the *CDC53mn* mutant is not caused by precocious SC disassembly. Based on the above results, we concluded that Cdc53 is necessary for efficient SC assembly.

We confirmed defective SC formation in the mutant by analyzing the localization of the chromosomal axis proteins, Hop1 (Fig 4A). In wild-type cells, Hop1 was initially bound to unsynapsed chromosomes as multiple foci/lines, and subsequently, large fractions of Hop1 dissociated from synapsed chromosomes (Fig 4B), as shown previously (Smith and Roeder, 1997; Bani Ismail et al, 2014). The *CDC53mn* mutant accumulated Hop1 on chromosomes as multiple foci (Fig 4A and B). Even at late time points (6 and 8 h), the multiple Hop1 foci/lines persisted on the chromosomes and disappeared only at later time points. The appearance and disappearance of Hop1 were delayed in the mutant by ~2 and ~4 h, respectively, relative to the wild type (Fig 4B).

We also examined the localization of other axis components, such as Rec8 and Red1 (Smith & Roeder, 1997; Klein et al, 1999). Wild-type cells showed dot/short line staining of Red1 and Rec8 in early prophase I. At the pachytene stage, when synapsis was almost complete, both proteins showed beads-in-line staining (Fig 4C), which was different from Hop1. In wild-type spreads, both Rec8 and

---

contain polycomplexes (PCs), as shown by an arrow. Bars = 1 μm. **(F)** Plots show each class of SC (wild type, left; *CDC53mn* mutant, right) at the indicated time points during meiosis. Class I (dots; blue bars), Zip1 dots; Class II (short lines; green bars), partial Zip1 linear; Class III (long lines; red bars), linear Zip1 staining. At each time point, more than 100 cells were counted. The representative results (*n* = 3) are shown; see the same results shown in Figs 3H and 5C. **(G)** The kinetics of Zip1 PC formation is shown for each strain. Spreads with Zip1 PC were counted. Wild type, blue; *CDC53mn*, red. Representative results are shown (*n* = 3); see the same results shown in Fig 5D. **(H)** Kinetics of Rad51 assembly/disassembly. The number of Rad51-positive cells (with more than five foci) was counted at each time point. At each time point, more than 100 cells were counted. Representative results are shown (*n* = 3); see the same results shown in Figs 2J, 3D, and 5D. Wild type, blue; *CDC53mn*, red.
Source data are available for this figure.

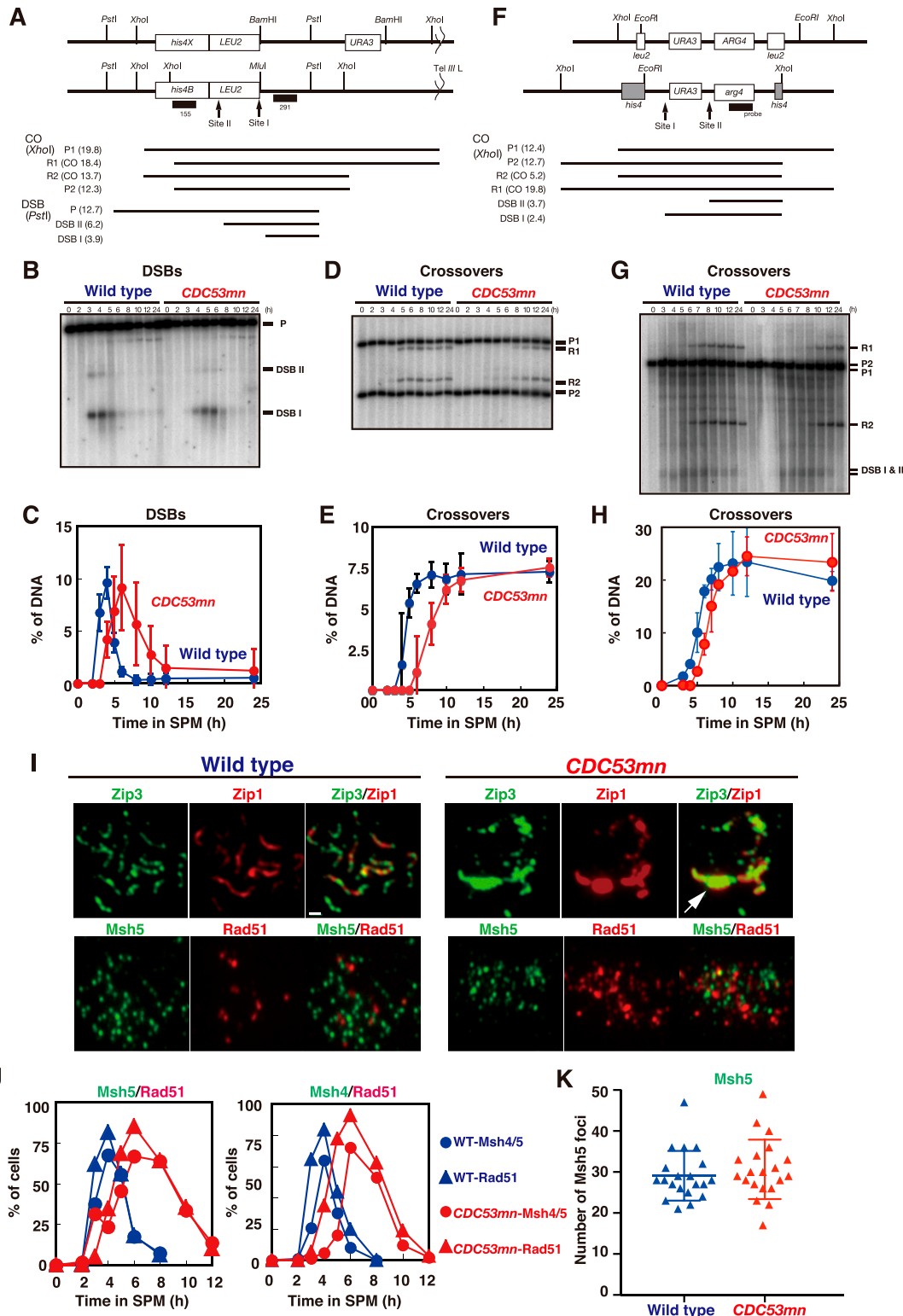

**Figure 2. Cdc53-depletion mutant is proficient in meiotic recombination.**
**(A)** A schematic diagram of the *HIS4-LEU2* recombination hotspot. Restriction sites for *Pst*I, *Xho*I, *Bam*HI, and *Mlu*I are shown. Diagnostic fragments for analysis on double-strand break (DSB) and crossover (CO) are shown at the bottom. The sizes of each fragment (kilo-bases) are presented within parentheses. **(B, C)** DSB repair at the *HIS4-LEU2* locus was analyzed by Southern blotting (B) and quantified (C). Genomic DNA was prepared and digested with *Pst*I. Error bars (SD) are based on three independent experiments. Wild type, NKY1551, blue circles; *CDC53mn*, ZHY94, red circles. **(D, E)** CO formation at the *HIS4-LEU2* locus was analyzed by Southern blotting (D) and quantified (E). Ratios of the R1 band to P1 were calculated. Genomic DNA was digested with *Xho*I. Error bars (SD) are based on three independent time courses.

Red1 showed dots at 2 h and short and long lines with beads-in-line staining at 4 h (Fig 4C). In the wild type, Red1 and Rec8 disappeared at 6 h (Fig 4E). On the other hand, the *CDC53mn* mutant showed a few formations of Red1/Rec8 long beads-in-lines at all time points during prophase I (Fig 4C and E), consistent with SC defects in the mutant. Moreover, in the *CDC53mn* mutant, Red1 dots appeared from 2 h, short-line staining peaked at 6 h, and from 8 h, Red1 signals gradually decreased during further incubation (Fig 4E). Contrary to Red1, few Rec8 dot-positive spreads were observed in the mutant at 2 h. Short lines of Rec8 appeared from 4 h and peaked at 6 h, and some fractions of Rec8 short-lines started to disappear after 8 h, probably because of the cleavage-independent cohesin release (Challa et al, 2019). Rec8 dots persisted on chromosomes during further incubation because of metaphase-I arrest; Rec8 showed a little cleavage. This result indicated the uncoordinated loading of axis proteins Red1 and Rec8/Hop1 during early meiosis in the *CDC53mn* mutant. Indeed, the mutant showed spreads at early time points such as 3 h, which were positive for Red1 but negative for Rec8 (Fig 4D). However, this staining was rarely observed in early wild-type spreads (Fig 4C). Rather than delayed loading of Hop1/Rec8, the mutant exhibited precocious loading of Red1 and Zip1 (Fig 1F), supporting the uncoordinated loading of chromosomal proteins. These results imply that the *CDC53mn* mutant assembles an altered structure of the meiotic chromosome axis.

We also analyzed the chromosome axis structure by deconvolution analysis of DAPI-stained chromosomes. In the wild type, two pairs of DAPI-stained lines were often co-aligned with each other (Fig 5A). Like the *zip3* mutant (Agarwal & Roeder, 2000), the *CDC53mn* mutant formed few DAPI-dense linear structures. Thus, these results suggest that Cdc53 might promote the proper formation of chromosome axes. To probe the chromosome axis structure, we compared Rec8 staining in the *CDC53mn* mutant with those in the *zip3* and *gmc2* mutants, both of which were defective in SC formation. In the *zip3* mutant, which shows zippering-defect of SCs (Agarwal & Roeder, 2000), beads-on-line staining of both Rec8 and Red1 was observed at late time points, suggesting normal assembly of AE or chromosome axis in the *zip3* mutant. On the other hand, the *CDC53mn* mutant showed more dotty staining of Rec8 than *zip3* (Fig 3I). The *gmc2* mutant is also defective in SC elongation but retains the ability to form COs (Humphryes et al, 2013), which is similar to the *CDC53mn* single mutant. Unlike the *CDC53mn* mutant, the *gmc2* mutant also exhibited elongated Rec8 lines (Fig S3E). These results indicate that the *CDC53mn* mutant forms an altered chromosome axis, which is different from other SC-defective mutants.

During meiotic prophase I, chromosome axes are compacted compared with premeiotic cells (Challa et al, 2016, 2019). We measured

the distance between CenIV and TelIV loci marked with GFP on chromosome IV in an intact cell (Figs 4F and G and S2C). The distance between the two loci in wild-type cells at 4 h was 0.89 ± 0.26 µm (*n* = 166), whereas that in *CDC53mn* mutant cells at 6 h was 1.06 ± 0.35 µm (*n* = 166, Mann–Whitney *U* test, *P*-value < 0.0001). This showed that the *CDC53mn* mutant is weakly defective in chromosome compaction during meiosis.

### Deletion of *PCH2* largely suppresses SC defect by Cdc53 depletion

As an SCF component, Cdc53 facilitates the degradation of a target protein Sic1, which functions as a negative regulator for biological processes. We hypothesized that Cdc53 might relieve the negative regulation of SC formation and looked for a mutant that rescues a defect in the *CDC53mn* mutant. We found that the deletion of the *PCH2* gene largely suppressed the SC defects induced by Cdc53 depletion (Fig 5A and B). The *pch2* mutation was originally isolated as a suppressor mutation that alleviates meiotic prophase arrest induced by the *zip1* mutation (San-Segundo & Roeder, 1999). Pch2 is a conserved AAA+ ATPase, which is associated with various cellular activities (Wu & Burgess, 2006). Pch2 is involved in remodeling of chromosome axes by modulating Hop1 (Borner et al, 2008; Chen et al, 2014), as well as in the pachytene checkpoint and regulation of DSB formation (Wu & Burgess, 2006). As described previously (San-Segundo & Roeder, 1999; Borner et al, 2008), the *pch2* single mutant showed more continuous Zip1 lines than the wild type (Fig 5A and B) (Borner et al, 2008). In contrast to the *CDC53mn* single mutant, the *CDC53mn* mutant with the *pch2* deletion formed uniformly stained long Zip1 lines like the *pch2* single mutant (Fig 5A and B). Staining of Rec8 revealed Rec8 lines in the *CDC53mn pch2* double mutant, similarly to the *pch2* single mutant (Fig 5B), suggesting that *pch2* suppresses the SC defect in the *CDC53mn* mutant.

We checked whether SCs in the double mutant were formed between homologous or non-homologous chromosomes. The pairing of a centromere locus was tested using CenXV-GFP (Fig S4D). Thirty percent of the wild-type spreads (n = 50) harboring one spot peaked at 5 h, probably because of the transient nature of pairing at the locus. *CDC53mn* showed a similar frequency of 28% (at 6 h), indicating that the mutant is proficient in the pairing. The *CDC53mn pch2* double mutant accumulated spreads with one spot with a frequency of 74% at 8 h, indicating a normal pairing of CenXV-GFP in the double mutant. These results indicate that SCs in the double mutant are formed between homologous chromosomes.

The *CDC53mn pch2* double mutant initially accumulated Zip1 PCs with defective SC assembly at earlier time points as seen in the

---

**(F)** A schematic diagram of the *URA3-ARG4* recombination hotspot. Restriction sites for *Xho*I and *EcoR*I are shown. Diagnostic fragments for analysis of parent, DSB, and crossover (CO) fragments are shown at the bottom. The sizes of each fragment (kilo-bases) are presented within parentheses. **(G, H)** CO formation in the *CDC53mn* mutant was verified by Southern blotting. **(G, H)** Ectopic CO formation at the *URA3-ARG4* recombination locus was analyzed by Southern blotting (G) and quantified (H). Genomic DNA was digested with *Xho*I. Error bars (SD) are based on three independent cultures. Wild type, MJL2442, blue circles; *CDC53mn*, ASY1202, red circles. **(I)** The localization of Zip3 and Msh5 in the *CDC53mn* mutant. Chromosome spreads from wild type (4 h, NKY1551) and *CDC53mn* mutant (8 h, ZHY94) were stained with anti-Zip3 or anti-Msh5 antibodies (green) together with anti-Zip1 (red). The representative images are shown. Bar = 1 µm. **(J)** The assembly of Msh4-Msh5 in the *CDC53mn* mutant. The percentages of cells positive for Msh4, Msh5, or Rad51 foci (more than five foci per nucleus) were counted at each time point. At least 100 nuclei were counted at each time point. Wild type, blue triangles and circles; *CDC53mn* mutant, red triangles and circles. Triangles and circles are for Rad51 and Msh4 (right) or Msh5 (left), respectively. The representative results are shown (*n* = 2). **(K)** The number of Msh5 foci per spread was counted at 4 h in the wild type (NKY1551) and at 6 h in the *CDC53mn* mutant (ZHY94). Twenty spreads were counted. The mean and SD are shown in the plot.
Source data are available for this figure.

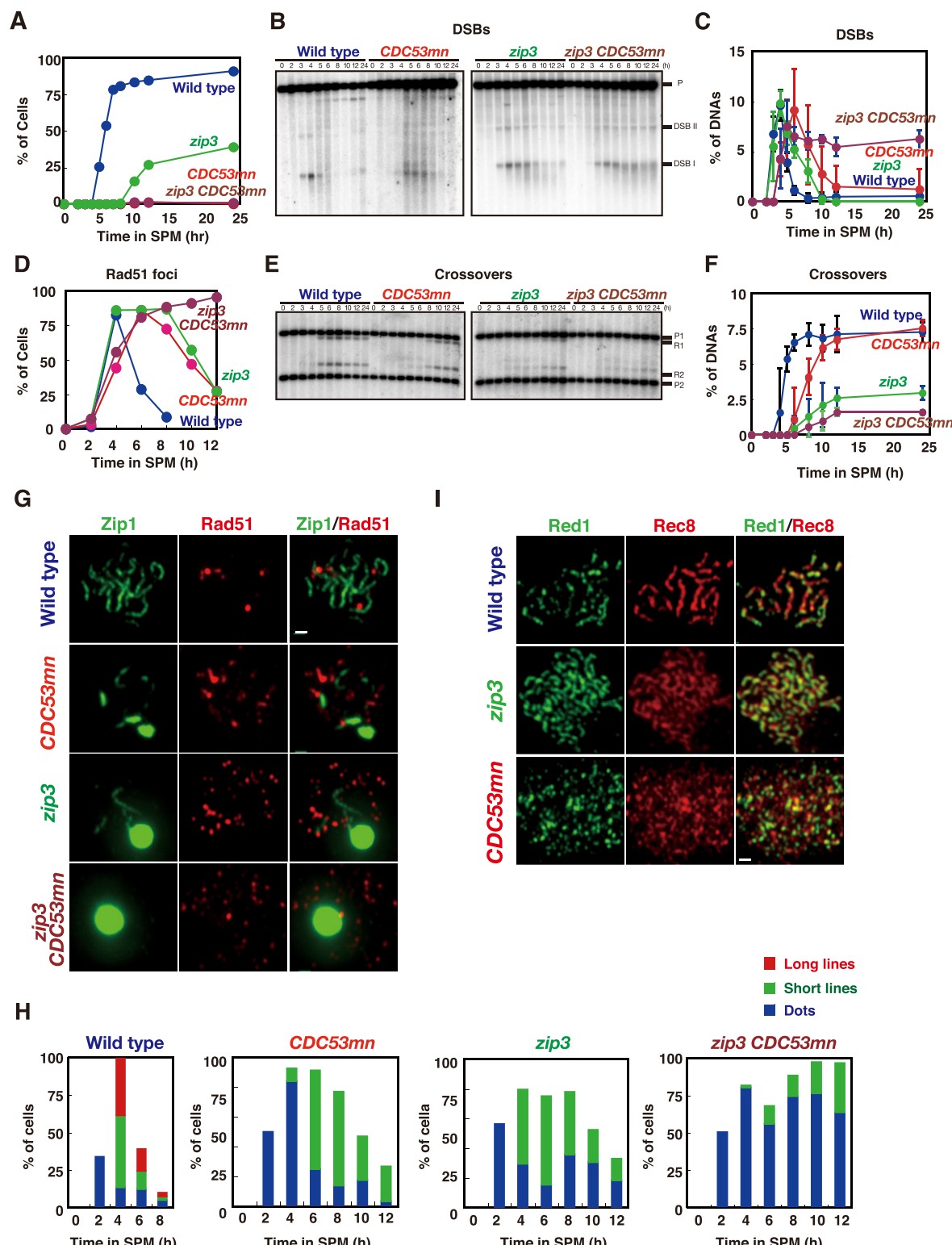

**Figure 3. Cdc53 and Zip3 work independently in synaptonemal complex formation and meiotic recombination.**
**(A)** The cell cycle progression of various mutants. The entry into meiosis I in the wild-type (NKY1551, blue), *CDC53mn* (ZHY94, red), *zip3* (MSY2889, green), *zip3 CDC53mn* mutant (ZHY259, brown) cells were analyzed by DAPI staining/counting as described in Fig 1B. Representative results are shown (n = 3). **(B, C)** Double-strand break repair at the *HIS4-LEU2* locus in various strains was analyzed as described above. Blots (B) and quantifications (C) are shown. Error bars (SD) are based on three independent cultures. **(D)** Rad51 staining in various mutants was analyzed as described above. **(G)** Typical staining patterns of each mutant are shown in (G). At least 100 spreads were counted at each time point. Wild type, blue; *CDC53mn*, red; *zip3*, green; *zip3 CDC53mn*, brown. **(E, F)** CO formation in various strains was analyzed at the *HIS4-LEU2* locus, as

*CDC53mn* single mutant; however, during further incubation, the PCs disappeared, and concomitantly full-length SCs appeared in the double mutant (Fig 5C and D). At 8 h, most of the nuclei of the double mutant contained full SCs without any PCs. These results suggest that suppression of SC defects by the *pch2* mutation is not due to suppression of early defects conferred by Cdc53 depletion. Consistent with this, neither delayed onset of S phase nor delayed degradation of Sic1 and Cdc6 in *CDC53mn* cells was suppressed by *pch2* (Fig S4A and B). The effect of *pch2* deletion on the suppression of SC defects is highly specific to Cdc53-deficiency in SC assembly because the *pch2* deletion mutation does not suppress SC defects induced by the *zmm* mutation (San-Segundo & Roeder, 1999).

Although the *CDC53mn pch2* double mutant appeared to form normal SCs, the mutant was arrested at the pachytene stage with full-length SCs and did not disassemble SCs or exit this stage (Figs 5C and 6A). We next checked the meiotic recombination. The *CDC53mn pch2* double mutant could not repair meiotic DSBs at the *HIS4-LEU2* locus and accumulated more processed DSB ends (Fig 6B and C), which was accompanied by the accumulation of Rad51 foci on spreads (Fig 6D). Consistent with the DSB repair defect, the *CDC53mn pch2* double mutant showed largely decreased CO levels compared with the wild-type strain (~1/5 of the wild type level; Fig 6E and F). On the other hand, the *CDC53mn* (Fig 3) and *pch2* single (Hochwagen et al, 2005; Borner et al, 2008) mutants showed weak DSB repair defects with normal CO formation (Fig 6B–F). We also checked genome-wide DSB repair by examining the chromosome bands using PFGE (Fig S1E). Unlike the *CDC53mn* and *pch2* single mutants, *CDC53mn pch2* did not recover the bands of intact chromosomes at late time points, indicating little DSB repair. These results indicate that Pch2 and Cdc53 work distinctly for meiotic DSB repair and/or CO formation. Furthermore, this implies that the completion of the recombination (DSB repair) is not necessary for the formation of full SCs/synapsis. As pointed out by Kleckner et al (1991), early recombination events are likely sufficient to promote synapsis between homologous chromosomes (see the Discussion section). Consistent with the arrest with unrepaired DSBs, the *CDC53mn pch2* double mutant did not express the mid-pachytene exit marker Cdc5 (Fig S4B). The mid-pachytene arrest in the double mutant is induced by the recombination checkpoint during meiosis (Hollingsworth & Gaglione, 2019).

Because Pch2 regulates the localization of Hop1 on chromosomes (Borner et al, 2008), we checked the expression and localization of the Hop1 protein (Hollingsworth et al, 1990). In the wild type, Hop1 protein was induced after entry into meiosis and persisted during meiosis (Fig S4C). Hop1 showed band shifts from 3 h by phosphorylation in a Mec1 (ATR) kinase–dependent manner (Carballo et al, 2008). These phosphorylated bands of Hop1 decreased during late prophase I of the wild type. The *CDC53mn* mutant induced similar amounts of Hop1 protein and phosphorylated Hop1 compared with the wild type,

although the appearance of phospho-Hop1 was delayed by ~2 h due to delayed DSB formation (Fig 2B and C).

We checked the localization of Hop1 by immunostaining. In the wild type, Hop1 showed dotty or short-line staining before full synapsis, and most of Hop1 dissociated from the synaptic regions of pachytene chromosomes (Fig 6G). As a result, the Hop1 localization of pachytene chromosomes was largely decreased. As reported (Borner et al, 2008), the *pch2* mutant accumulated "unusual" long Hop1 lines during mid-prophase I, which colocalized with Zip1 lines. In the *CDC53mn pch2* mutant, as seen in the *pch2* mutant, Hop1 accumulated on full-length Zip1 lines (Fig 6G). These results indicate that, unlike Pch2, Cdc53 does not affect the Hop1 protein levels and its localization on chromosomes.

One possible explanation for the suppression of SC defects in the *CDC53mn* mutant by *pch2* is that Pch2 is a target of Cdc53-dependent protein degradation. To check this possibility, we analyzed the amounts of Pch2-HA protein by Western blotting. The wild-type cells showed an increase in Pch2 levels from 1 h of meiosis. Pch2 levels peaked at 4 h and disappeared at 6 h (Fig 6H). The levels of Pch2 protein in the *CDC53mn* mutant were similar to those in the wild type during meiosis. The amount of Pch2 decreased slightly at late time points. This suggests that the Pch2 protein level is not affected by Cdc53 depletion. There were a few band shifts of Pch2 during meiosis in the wild type and *CDC53mn* mutant strains by modifications (Fig 6H).

We checked Pch2 localization using an anti-Pch2 antibody. In wild-type cells, as shown previously (San-Segundo & Roeder, 1999), Pch2 localizes in both chromosomes and nucleolus (Fig 6I). On the *CDC53mn* spreads, strong signals of Pch2 were seen on Zip1 PCs. This is consistent with the accumulation of Pch2 on Zip1-PC in other synapsis-defective mutants (San-Segundo & Roeder, 1999; Herruzo et al, 2016). The *CDC53mn* cells also showed clustered foci on the nucleolus, although these Pch2 signal intensities were relatively weaker than those in the wild type (arrowheads in Fig 6I). Chromosomal Pch2 signals were much weaker in *CDC53mn* cells than in wild-type cells. Because Pch2 localization on chromosomes depends on Zip1 (San-Segundo & Roeder, 1999; Herruzo et al, 2016), the reduced loading of Pch2 in the mutant might be due to defective Zip1 elongation.

## The *cdc4* mutant shows a defect in SC assembly

We next looked for an F-box protein working with Cdc53 during meiosis. Since *CDC53mn* cells accumulate Cdc6, whose degradation during mitosis depends on Cdc4 (Perkins et al, 2001), we examined the role of Cdc4 in SC formation by using the temperature-sensitive *cdc4-3* mutant in vegetative growth. Previous analysis showed that this mutant accumulated Sic1 at 36°C with a delayed S-phase entry (Sedgwick et al, 2006). The *cdc4-3* mutant showed defective Zip1 elongation at 32°C, which is a semi-permissive temperature for the

described above. Blots (E) and quantifications (F) are shown. Error bars (SD) are based on three independent experiments. **(G, H)** The chromosome spreads from wild-type, *CDC53mn*, *zip3*, *zip3 CDC53mn* cells were stained with anti-Zip1 (green) as well as anti-Rad51 (red) antibodies, and the staining pattern for Zip1 was classified into classes and plotted (H) as shown in Fig 1F. The representative results are shown (*n* = 2). Bar = 1 μm. **(I)** The chromosome axis formation was analyzed by staining the chromosome spreads from various strains with anti-Red1 (green) and anti-Rec8 (red) antibodies. Wild type, 4 h; *CDC53mn*, 8 h; *zip3*, 8 h; *zip3 CDC53mn*, 8 h. Bar = 1 μm. Source data are available for this figure.

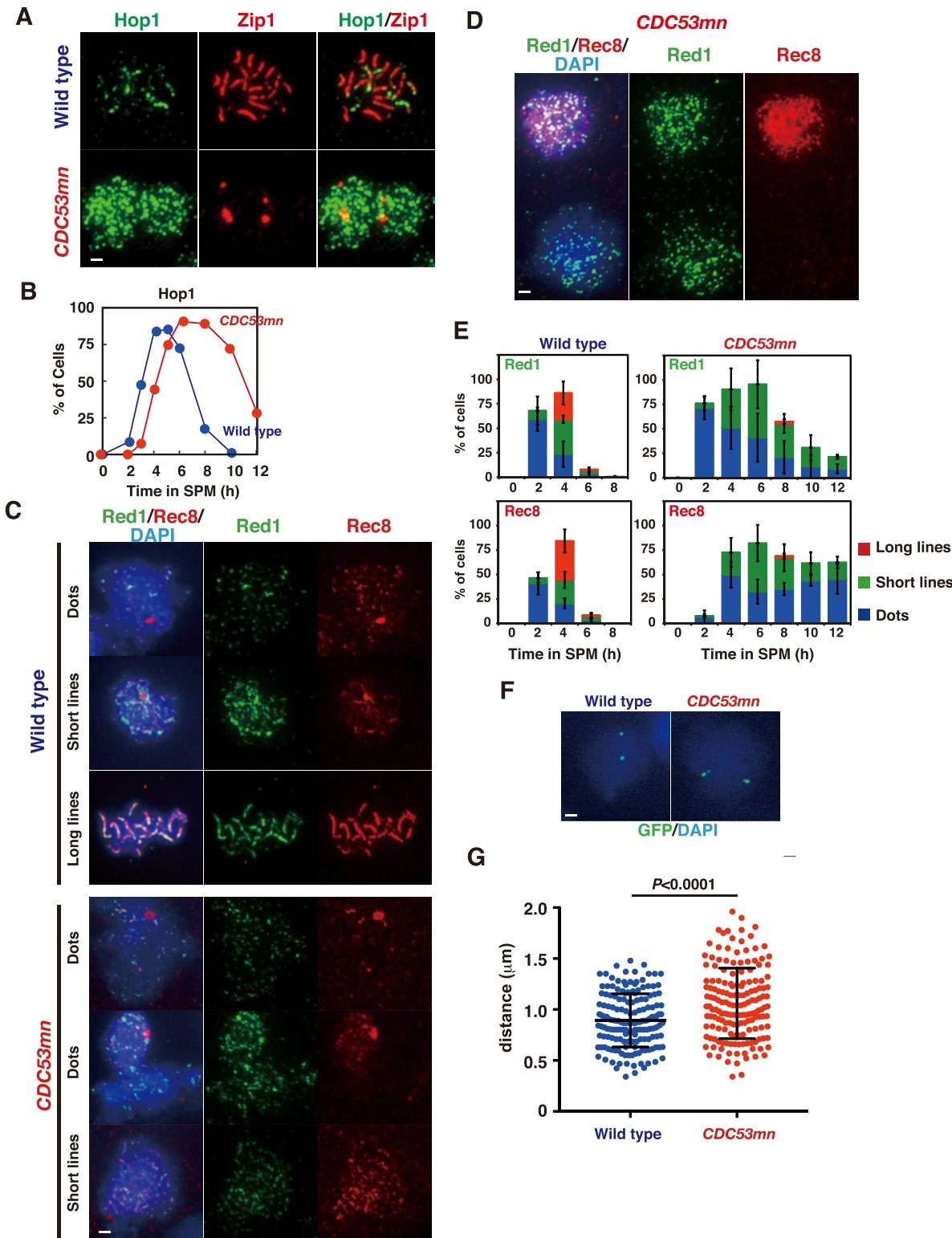

**Figure 4. Cdc53 depletion induces altered axis formation.**
**(A)** Hop1 staining in the wild type (NKY1551) and *CDC53mn* mutant (ZHY94). Chromosome spreads in each strain were stained with anti-Hop1 (green) and anti-Zip1 (red) antibodies. For the wild type, pachytene chromosomes with long Zip1 lines are shown with a few Hop1 foci. The representative images are shown. Bar = 1 $\mu$m. **(B)** The kinetics of Hop1 assembly/disassembly. The number of Hop1-positive cells was counted at each time point. More than 50 spreads were counted. The representative results are shown (n = 2). Wild type, blue circles; *CDC53mn*, red circles. **(C, D)** The nuclear spreads from the wild type and *CDC53mn* mutant were stained with anti-Red1 (green), anti-Rec8 (red), and DAPI (blue) and then categorized. Synaptonemal complexes of wild-type cells are shown as dots, short lines, and long lines. Synaptonemal

mutant, and accumulated Zip1-PCs (Fig 7A). Wild-type cells at 32°C exhibited similar Rad51- and Zip1-staining kinetics to those at 30°C (compare with Fig 1). The *cdc4-3* mutant showed normal Rad51-assembly and delayed Rad51-disassembly at 32°C (Fig 7B), implying the role of Cdc4 in meiotic DSB repair. More importantly, *pch2* deletion also suppressed the Zip1 assembly defect observed in the *cdc4-3* mutant at 32°C (Fig 7C–E). The *cdc4-3 pch2* double mutant formed long Zip1 lines like the *pch2* single mutant (Fig 7C). The *cdc4-3 pch2* mutant showed little SC disassembly even at late time points (Fig 7E). Most of the *cdc4-3* mutant cells did not show meiosis I arrest and showed a delayed entry into meiosis I at 32°C (Fig 7F). These results indicate that Cdc4 regulates SC formation together with Cdc53, but not at the onset of anaphase I.

# Discussion

SCF ubiquitin ligase complexes are involved in various cellular processes during mitosis and meiosis, particularly in cell cycle progression. In this study, by analyzing the role of two SCF components, Cdc53 (cullin) and Cdc4 (F-box protein) in meiotic cells, we showed that Cdc53 and Cdc4 promote SC formation and that Cdc53, but not Cdc4, is required for the transition from metaphase I to anaphase I. Given that Cdc53 as an SCF component mediates protein ubiquitylation, it is likely that SCF-dependent ubiquitylation is involved in the two critical meiotic events.

## The role of SCF in SC assembly

The role of ubiquitylation in meiotic chromosome metabolism during meiotic prophase I has recently been shown in a study of Hei10 ubiquitin ligase in mice (Qiao et al, 2014). Hei10 plays a direct role in meiotic recombination and an indirect role in SC formation by antagonizing the SUMO ligase Rnf212, a Zip3 ortholog of budding yeast (Qiao et al, 2014). Moreover, in budding yeast, nematodes, and mice, the proteasome localizes on meiotic chromosomes (Ahuja et al, 2017; Rao et al, 2017), supporting the role of protein ubiquitylation in meiotic chromosome functions. Our studies showed the role of SCF-dependent ubiquitylation in SC formation.

Like the *zmm* and *ecm11/gmc2* mutants (Humphryes et al, 2013; Pyatnitskaya et al, 2019), the *CDC53mn* mutant is deficient in SC elongation. Importantly, there are differences in defects in meiotic chromosomal events among these mutants. The *zmm* mutants are defective in CO formation, whereas the *ecm11/gmc2* and *CDC53mn* mutants are proficient. The *ecm11/gmc2* mutant showed normal assembly of ZMM proteins, including Zip3 and Msh4. On the other hand, the *CDC53mn* mutant was defective in the ZMM assembly,

except for Msh4-5. This suggests the involvement of Cdc53 in the Msh4/5-independent ZMM function (Shinohara et al, 2008; Pyatnitskaya et al, 2019), which might be unrelated to its role in CO formation.

In *S. cerevisiae*, the formation of AEs is coupled with SC elongation accompanied by the polymerization of transverse filaments (Padmore et al, 1991). Moreover, whereas the *zmm* and *ecm11/gmc2* mutants form normal chromosome axes (Humphryes et al, 2013; Pyatnitskaya et al, 2019), the *CDC53*-depletion mutant exhibits altered axis assembly. SCF$^{Cdc4}$ seemed to regulate chromosomal events during early prophase I when axial protein(s) are loaded. *CDC53* depletion seemed to cause uncoupling of the loading of SC components, Red1 (and Zip1), with that of Rec8 cohesin and Hop1 to chromosomes. In the wild type, the loading of these proteins occurred in early prophase I at a similar time. On the other hand, the *CDC53mn* mutant exhibited distinct loading timing between Rec8-Hop1 and Red1-Zip1. Given a delay in the premeiotic S phase in the *cdc53* mutant, rather than delayed loading of Rec8 and Hop1, promiscuous uncoupled loading of meiosis-specific components such as Red1 (and Zip1) might occur in the mutant. These results suggest that coordinated loading of different proteins onto chromosomes promotes proper axis formation, which in turn facilitates SC assembly. We propose that, by functioning in early prophase I, the SCF$^{Cdc4}$ complex controls the coordinated loading of axis proteins for proper axis formation, which is critical for coupling of axis assembly with SC elongation.

Although we could not detect any immune-staining signals of Cdc53 on meiotic chromosome spreads (unpublished results), recent studies on mouse spermatocytes showed the localization of an SCF component, Skp1, on the LE of SCs (Guan et al, 2020), which supports the role of SCF in axis assembly. It is known that SCF regulates mitotic chromosome condensation in fruit flies and nematodes (Feng et al, 1999; Buster et al, 2013). Recently, SCF ubiquitin ligase has been shown to promote meiotic chromosome pairing as well as entry into meiosis in *Caenorhabditis elegans* (Mohammad et al, 2018) and mice (Gray et al, 2020; Guan et al, 2020), which are shared defects with the yeast *CDC53*-depletion mutant. Therefore, SCF ubiquitin ligase regulates chromosome morphogenesis during mitotic and meiotic phases in various organisms.

## SC assembly is negatively regulated

Because the SCF complex promotes the ubiquitylation of a target protein for degradation, we postulated the presence of a negative regulatory pathway for SC assembly, which would be inactivated by the SCF (Fig 8). In the absence of SCF, the putative negative regulator might inhibit SC elongation. This putative negative regulator might be involved in proper coordination of axis assembly and SC

---

complexes in the *CDC53mn* mutants are shown as dots and short lines. **(D)** Staining of Rec8/Red1 in two adjacent chromosome spreads of the mutant at 3 h is shown. The top is positive for both Rec8 and Red1. The bottom is positive for Red1 but negative for Rec8. Bar = 1 $\mu$m. **(E)** Kinetics of each class of Red1 (upper two graphs) and Rec8 (lower two graphs) in the wild type and *CDC53mn* mutant at the indicated times during meiosis. Each stained image was classified into dots (blue bars), short lines (green bars), and long lines (red bars) and then counted. At each time point, more than 50 spreads were counted. Error bars (SD) indicate SD ($n$ = 4). **(F)** The chromosomal compaction was measured using cells with GFP-marked Cen*IV* and Tel*IV*. The representative images of the wild type and *CDC53mn* mutant are shown. Bar = 1 $\mu$m. **(G)** The distance between the two GFP signals was measured using NIH Image J and plotted. Wild type, ZHY749 (blue); *CDC53mn*, ZHY750 (red). Means ($n$ = 166) and SD are shown. The *P*-value was calculated using the Mann-Whitney *U* test.
Source data are available for this figure.

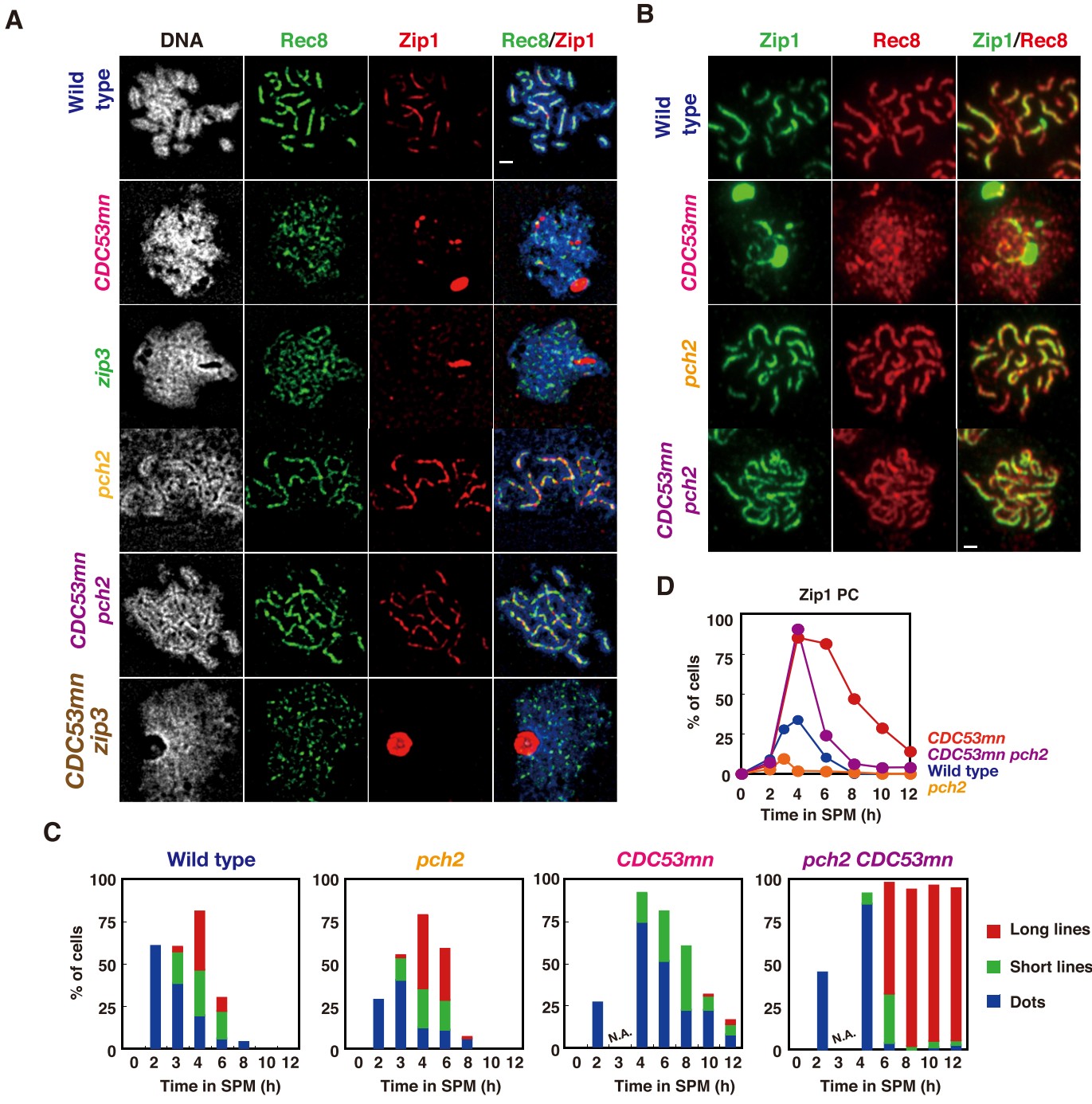

**Figure 5. The *pch2* mutation suppresses synaptonemal complex defects induced by Cdc53 depletion.**
**(A)** The chromosome spreads from various strains were stained with anti-Zip1 (red) and anti-Rec8 (green) antibodies as well as DAPI (white), and images were captured using DeltaVision epifluorescence microscope and deconvoluted as described in the Materials and Methods section. The representative images are shown. Wild type, NKY1551; *CDC53mn*, ZHY94; *zip3*, MSY2889; *pch2*, ZHY350; *pch2 CDC53mn*, ZHY351; *zip3 CDC53mn*, ZHY259. Bar = 1 μm. **(B)** The chromosome spreads from various strains were stained with anti-Rec8 (green) and anti-Zip1 (red) antibodies. Wild type, 4 h; *CDC53mn*, 8 h; *pch2*, 6 h; *pch2 CDC53mn*, 8 h. Bar = 1 μm. **(C)** Zip1-stained cells of each strain were classified and plotted at each time point. More than 100 nuclei were counted, as shown in Fig 1F. Class I (blue bars), Zip1 dots; Class II (green bars), partial Zip1 linear staining; Class III (red bars), linear Zip1 staining. N.A., not available. The representative results are shown (*n* = 2). **(D)** The cells containing Zip1 polycomplexes were counted at each time point and plotted. Wild type, blue; *CDC53mn*, red; *pch2*, orange; *pch2 CDC53mn*, purple. The representative results are shown (*n* = 2).
Source data are available for this figure.

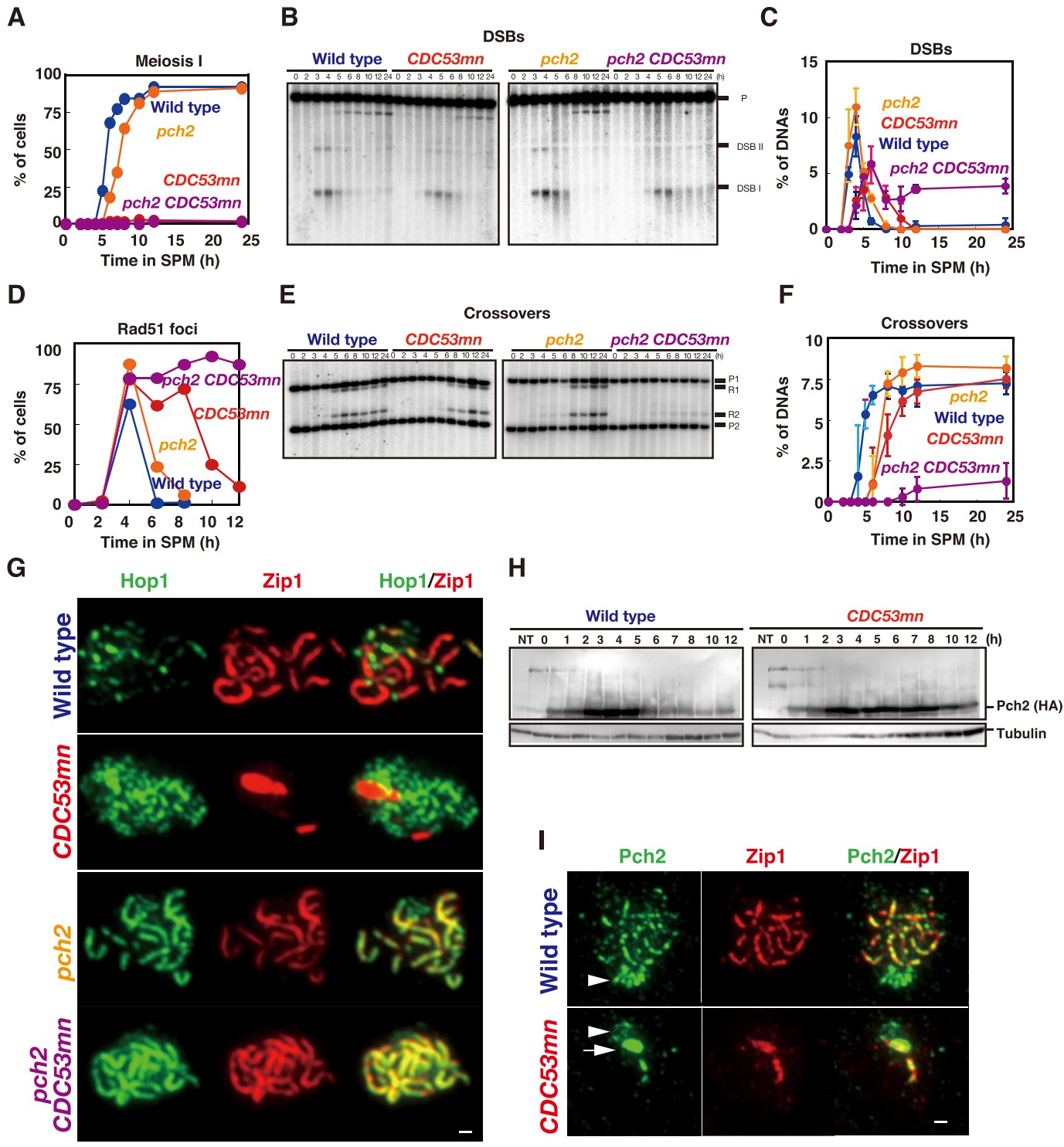

**Figure 6. The *pch2 CDC53mn* mutant is defective in meiotic recombination.**
**(A)** The entry into meiosis I of various strains was analyzed by DAPI staining. Wild type, NKY1551; *CDC53mn*, ZHY94; *pch2*, ZHY350; *pch2 CDC53mn*, ZHY351. **(B, C)** Double-strand break repair at the *HIS4-LEU2* locus in various strains was analyzed as described above. Blots (B) and quantifications (C) are shown. Error bars (SD) are based on three independent cultures. **(D)** The cells containing Rad51 foci were counted at each time point and plotted, as described above. The representative results are shown (*n* = 2). Wild type, blue; *CDC53mn*, red; *pch2*, orange; *pch2 CDC53mn*, purple. **(E, F)** CO formation in various strains was analyzed at the *HIS4-LEU2* locus as described above. Blots (E) and quantifications (F) are shown. Error bars (SD) are based on three independent time courses. **(G)** Hop1/Zip1 staining in various mutants. Chromosome spreads in each strain were stained with anti-Hop1 (green) and anti-Zip1 (green) antibodies. **(H)** The expression of Pch2-HA during meiosis. Cell lysates from *PCH2-3HA* and *CDC53mn PCH2-3HA* cells were analyzed by Western blotting using anti-HA (top) and anti-tubulin (bottom) antibodies. The representative results are shown (*n* = 2). **(I)** The chromosome spreads from various strains were stained with anti-Pch2 (green) and anti-Zip1 (red) antibodies. Wild type, 4 h; *CDC53mn*, 8 h. Arrows indicate the "polycomplex" and arrowheads indicate the nucleolus. Bar = 1 *μ*m.
Source data are available for this figure.

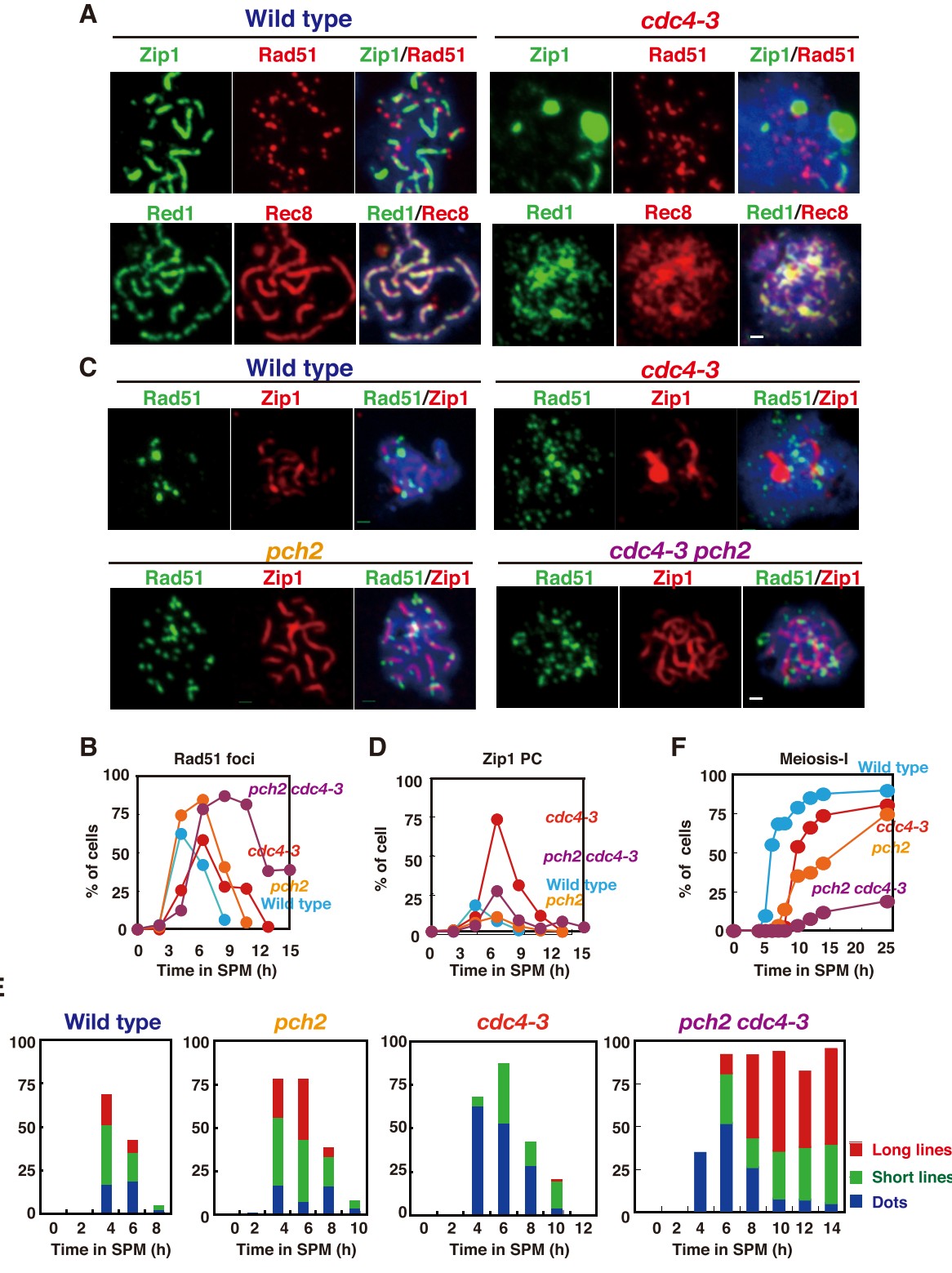

**Figure 7. The *cdc4-3* is defective in synaptonemal complex-assembly.**
**(A, B)** The chromosome spreads from wild-type and *cdc4-3* mutant cells incubated in SPM at 32°C were stained with anti-Zip1 (green, upper panels) and anti-Rad51 (red, upper panels), and with anti-Red1 (green, bottom panels) and anti-Rec8 (red, bottom panels), respectively, as described above. Wild type (NKY1551), 5 h; *cdc4-3* (ZHY522), 8 h. **(B)** Kinetics of Rad51 focus-positive cells are shown in (B). The representative results are shown (*n* = 2). Bar = 1 μm. **(C, D, E)** The chromosome spreads from various strains at 32°C were stained with anti-Zip1 (red) and anti-Rad51 (green) antibodies. Typical staining patterns are shown in (C). Wild type (NKY1551), 5 h; *cdc4-3* (ZHY522) 8 h; *pch2* (ZHY350) 8 h;

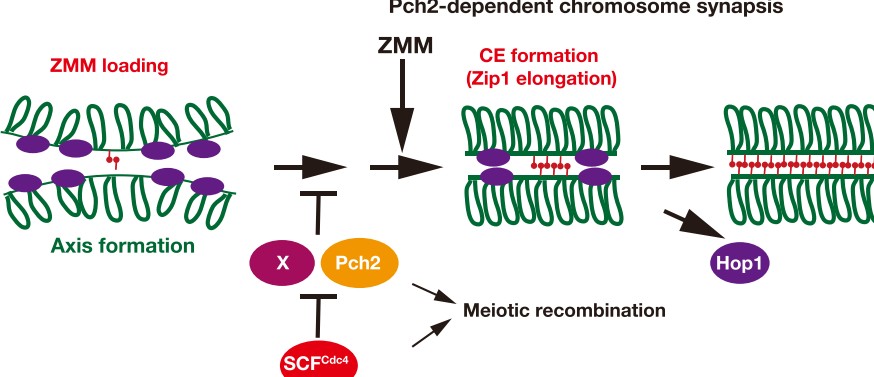

**Figure 8. A model showing the regulation of synaptonemal complex (SC) formation by SCF and Pch2.**
Refer to the text for more details. The SCF[Cdc4] promotes SC formation by down-regulating Pch2 and protein "X," both of which may negatively regulate proper axis assembly for SC formation. SCF[Cdc4] may promote the ubiquitylation of protein "X" for degradation to promote SC assembly.

elongation. In this scenario, we could expect to identify a mutation that suppresses the SC defect induced by *CDC53* depletion. Indeed, we found that the deletion of the *PCH2* gene suppresses a defect in SC assembly in *CDC53mn* and *cdc4* mutant cells. This suggests the presence of a Pch2-dependent negative regulation for SC assembly (Fig 8). One possibility is that Pch2 could be a direct target of SCF[Cdc4]-mediated ubiquitylation. However, contrary to this expectation, our study did not provide any evidence to support the hypothesis that Cdc53 controls the posttranslational status of Pch2. Thus, there might be another target of SCF[Cdc4] in SC assembly ("X" in Fig 8).

If Pch2 is not a substrate of SCF[Cdc4], how does *pch2* deletion suppress SC-assembly defects in the *CDC53mn* mutant? In SC assembly, Pch2 (TRIP13 in mammals) regulates the dissociation of a chromosome axis protein, Hop1 (HORMAD1/2 in mammals), which might control synapsis (Borner et al, 2008; Wojtasz et al, 2009). One likely possibility is that SCF[Cdc4] might control the Hop1-mediated regulatory pathway for SC assembly. A recent study showed that Pch2 controls the amount of Hop1 protein by modulating the conversion of a closed form of Hop1 to an active form (Raina & Vader, 2020). In the *pch2* mutant, Hop1 (closed form), which promotes chromosome synapsis, accumulated on chromosomes. In this scenario, increased levels of Hop1 on chromosomes due to *PCH2* depletion could indirectly suppress SC defects in the *CDC53mn* mutant.

Interestingly, the *CDC53*-depletion mutant showed wild-type levels of Hop1 protein with normal phosphorylation and normal loading (and unloading) of Hop1 on chromosomes (Fig 6). Therefore, it is very unlikely that the SCF directly down-regulates Hop1. Rather, we propose that, in the absence of Pch2, cells do not require SCF-dependent control for SC assembly. In other words, the *pch2* mutant cells could form SCs in the presence of a negative regulator for SC assembly (Fig 8). Pch2 may activate the negative regulator for its action, as seen in the activation role of TRIP13 in the spindle assembly checkpoint (Vader, 2015). Alternatively, Pch2 may impose a kinetic barrier for SC formation, which might function in parallel with the Cdc53-dependent pathway. The latter role has been proposed for the nematode Pch2 ortholog (PCH-2) for chromosome pairing and synapsis (Deshong et al, 2014).

In mouse Skp1 conditional knockdown spermatocytes, chromosome synapsis is partly defective with the accumulation of Hormad1/2 proteins on synaptic SC regions, suggesting premature SC disassembly (Guan et al, 2020). In the yeast *CDC53*-depletion mutant, SC formation was almost defective with the accumulation of the Hop1 protein. The *ndt80* mutation, which induced pachytene arrest, did not fully suppress SC-defects in *CDC53mn* mutant cells, arguing against premature SC disassembly in the yeast mutant.

### The role of SCF in meiotic recombination

In mice, SUMO-ligase Rnf212 and ubiquitin ligase Hei10 collaborate in meiotic recombination (Qiao et al, 2014). Budding yeast has an Rnf212 ortholog, Zip3, but does not have a Hei10 ortholog. In this study, we found that SCF ubiquitin ligase plays a role not only in SC formation but also in meiotic recombination. Rather than cooperation, Zip3 and Cdc53 distinctly control CO formation because the *zip3 CDC53mn* double mutant is more deficient in CO formation than the *zip3* single mutant. Moreover, Cdc53 is essential for CO formation in the absence of Pch2, which plays a weak role in recombination in the wild-type background (Borner et al, 2008). These results suggest a role for Cdc53 in meiotic recombination. Our results support the notion that not only SUMOlyation but also ubiquitylation plays a role in recombination during yeast meiosis. This is consistent with the result that deletion of the proteasome component Pre5 impairs meiotic recombination (Ahuja et al, 2017).

### Relationship between meiotic recombination and SC formation

Previous studies have shown a close association or coupling between meiotic recombination and SC formation. It is believed that SC regulates meiotic CO formation (Cahoon & Hawley, 2016; Gao &

*cdc4-3 pch2* (ZHY580) 8 h. Bar = 1 μm. Kinetics of poly-complex (D), as well as Zip-classes (E), were analyzed as described above. The representative results are shown (*n* = 2). **(F)** Entry into meiosis I in different strains was analyzed by DAPI staining as described above. The representative results are shown (*n* = 2). Source data are available for this figure.

Colaiacovo, 2018). Moreover, meiotic recombination promotes SC formation (Padmore et al, 1991; Kleckner, 2006). On the other hand, our studies revealed two extreme mutant situations: uncoupled meiotic recombination and SC elongation. First, in the case of the *CDC53mn* mutant, meiotic CO formed efficiently in the absence of fully elongated SCs. This indicates that SC, at least SC elongation, is not required for efficient formation of COs per se. However, we do not deny the possibility that a short stretch of SCs formed in the *CDC53mn* mutant was sufficient for CO formation. We also observed the wild-type number of Msh5 foci, which is likely to exhibit non-random distribution like the Zip3 foci (Fung et al, 2004; Zhang et al, 2014a; Zhang et al, 2014b), on meiotic chromosomes in the mutant. Although we did not detect other visible ZMM foci in the *CDC53mn* mutant; ZMM-dependent CO formation was functional in the absence of Cdc53 because the COs in the *CDC53mn* mutant still require Zip3 (Fig 3). This suggests that normal establishment of ZMM-dependent CO formation may occur in the absence of *CDC53*, thus indicating SC formation/proper axis assembly. On the other hand, we did not address the effect of *CDC53* depletion on the implementation (and/or maintenance) of CO control.

Second, in the *CDC53mn pch2* double mutant, we observed normal SC formation with little CO formation or DSB repair. This suggests that the formation of meiotic recombination products such as COs is not necessary for SC formation. Because SC assembly depends on DSB formation, as suggested previously (Kleckner, 2006), early recombination intermediates such as DSBs and/or single-stranded DNAs at recombination sites are sufficient to trigger SC assembly. Alternatively, similar to fruit flies and nematodes (Dernburg et al, 1998; McKim & Hayashi-Hagihara, 1998), the *CDC53mn pch2* double mutant may induce an alternative SC-assembly pathway, which is less dependent on recombination.

### Implication for pachytene checkpoint

The *CDC53mn* mutant, which is defective in SC formation, but is proficient in meiotic recombination, passes through the pachytene stage and proceeds at least to metaphase-anaphase I transition. Indeed, the *CDC53mn* mutant expresses Cdc5 and Clb1, which are under the control of the Ndt80-dependent mid-pachytene checkpoint. This strongly suggests that abnormal SC in the mutant does not trigger any delay or arrest in meiotic prophase I, suggesting the absence of a checkpoint, which monitors synapsis (SC elongation, long SCs) under these conditions. Alternatively, Cdc53 might mediate synapsis checkpoint signaling.

Pch2 is involved in the mid-pachytene checkpoint (San-Segundo & Roeder, 1999). In this study, we found that the *pch2* mutation, when combined with *CDC53mn*, did not accelerate cell cycle progression, but showed an arrest in the mid-pachytene stage due to the inability to repair DSBs. This supports the role of Pch2 in meiotic recombination but not in the recombination checkpoint. Consistent with this, a mutation of the Pch2 homolog Trip13 in mice did not alleviate any defects in various meiotic recombination and synapsis-defective strains (Li & Schimenti, 2007). In addition, plants do not seem to show synapsis checkpoint during meiosis since meiotic cells progress through the cell cycle even in the presence of defective synapsis (Hamant et al, 2006).

### The role of SCF in metaphase I to anaphase I transition

In addition to roles during prophase I, SCF might be involved in the transition from metaphase I to anaphase I. The simplest interpretation is that SCF may degrade an inhibitor molecule for APC/C at the transition. The *cdc4* mutant is deficient in SC formation but is proficient in the transition from metaphase I to anaphase I, suggesting the involvement of an F-box protein other than Cdc4 in this transition. Similar arrest at metaphase I/anaphase I transition was reported in a yeast mutant of *RAD6* (Yamashita et al, 2004), which encodes an E2 enzyme for ubiquitylation. The role of Rad6 in SC formation is less clear because the *rad6* mutant also reduces DSB formation, which also affects SC assembly. Moreover, at metaphase/anaphase II transition in *Xenopus* oocytes, SCF is known to promote the degradation of an APC/C inhibitor, Erp1/Emi2 (Nishiyama et al, 2007). We need to identify a target molecule of the SCF ubiquitin ligase not only for SC assembly but also for the onset of anaphase I. Indeed, two recent studies have shown the role of SCF in the transition in mouse meiosis. One study showed the role of SCF in the activation of Wee1 kinase, which negatively regulates cyclin-dependent kinase (Cdk) in anaphase I onset (Gray et al, 2020). The other study showed the role of SCF in MPF (Cdk) activation in both spermatocytes and oocytes (Guan et al, 2020). The role of SCF in the metaphase I to anaphase I transition seems to be conserved from yeast to higher eukaryotes.

## Materials and Methods

### Strains and plasmids

All strains described here are derivatives of SK1 diploids, NKY1551 (*MATα/MATa, lys2/", ura3/", leu2::hisG/", his4X-LEU2-URA3/his4B-LEU2,* and *arg4-nsp/arg4-bgl*) except *cdc4-3* strain, which is a congenic strain. An ectopic recombination system with the *URA3-ARG4* cassette was provided by Dr. Michael Lichten. SK1 *cdc4-3* strain and *CENXV-GFP* strains were a kind gift from Drs. D Stuart and D Koshland, respectively. The genotypes of each strain used in this study are described in Table S1.

### Strain construction

*pCLB2-3HA-CDC53* were constructed by replacing an endogenous promoter with the promoter from the *CLB2* gene. The addition of the HA tag is important to deplete Cdc53 during meiosis. *pch2, zip3,* and *ndt80* null alleles were constructed by PCR-mediated gene disruption using either the *TRP1* or *LEU2* genes (Wach et al, 1994). *REC8-3HA, PCH2-3Flag* (*-3HA*), and *CDC53-3Flag* were constructed by a PCR-based tagging methodology (De Antoni & Gallwitz, 2000).

### Anti-serum and antibodies

Anti-HA antibody (16B12; Babco), anti-Flag (M2; Sigma-Aldrich), anti-tubulin (MCA77G; Bio-Rad/Serotec, Ltd), anti-GFP (3 × 10$^6$; Molecular Probes), and guinea pig anti-Rad51 (Shinohara et al, 2000) were used for staining. Anti-Cdc53 is a generous gift from Dr. M Blobel.

Anti-Zip1, -Zip3, -Zip2, -Mer3, -Spo22/Zip4, -Msh4, and -Msh5 as well as anti-Red1 were described previously (Shinohara et al, 2008). Anti-Rec8 antibody was described previously (Rao et al, 2011). Anti-Hop1 serum was described in Bani Ismail et al (2014). Anti-Sic1 (sc-50441, 1:1,000) and anti-Cdc5 (sc-33635, 1:1,000) antibodies were purchased from SantaCruz Biotech. Anti-Cdc6 (Cdc6 9H8/5) was purchased from Abcam. The second antibodies for staining were Alexa 488 (Goat) and 594 (Goat) IgG used at a 1/2,000 dilution (Thermo Fisher Scientific).

Anti-Pch2 was raised against recombinant N terminus 300 amino acid of truncated protein purified from *Escherichia coli*. An open-reading frame of the truncated Pch2 was PCR-amplified and inserted into pET15b plasmid (Novagen) in which an N terminus of the *PCH2* gene was tagged with hexa-histidine. Pch2 protein with the histidine tag was affinity-purified using the Nickel resin as described by manufactures and used for immunization (MBL Co. Ltd).

For double staining the following combinations were used; anti-Rad51 (guinea pig) and anti-Zip1 (rabbit), anti-ZMM (Zip2, Zip3, Zip4/Spo22, Msh4, Msh5, all rabbit) and anti-Zip1 (rat); anti-Red1 (chicken), anti-Rec8 (rabbit); anti-Zip1 (rat), anti-Rec8 (rabbit); anti-Zip1 (rat), anti-Pch2 (rabbit); anti-Zip1 (rat), anti-Hop1 (rabbit).

## Cytology

Immunostaining of chromosome spreads was performed as described previously (Shinohara et al, 2000, 2003). Stained samples were observed using an epi-fluorescent microscope (BX51; Olympus) with a 100 X objective (NA1.3). Images were captured by CCD camera (Cool Snap; Roper) and, then processed using IP lab and/or iVision (Sillicon), and Photoshop (Adobe) software. For focus counting, more than 100 nuclei were counted at each time point. Zip1 PCs were defined as a relatively large Zip1 staining outside of the DAPI staining region.

High-resolution images were captured by a computer-assisted fluorescence microscope system (DeltaVision; Applied Precision). The objective lens was an oil-immersion lens (100X, NA = 1.35). Image deconvolution was carried out using an image workstation (SoftWorks; Applied Precision).

Pairing of chromosomes was analyzed in the whole yeast cells with two homologous LacI-GFP spots at *CENX* locus. Following fluorescence microscope imaging, the number of chromosomal locus-marked GFP foci in a single cell was counted manually.

The distance between two GFP foci on chromosome IV on a single focal plane in a intact yeast cell taken by the fluorescence microscope system (DeltaVision; Applied Precision) was measured by NIH image J program.

## Western blotting

Western blotting was performed as described previously (Hayase et al, 2004; Shinohara et al, 2008). Western blotting was performed for cell lysates extracted by the TCA method. After being harvested and washed twice with 20% TCA, the cells were roughly disrupted by Yasui Kikai (Yasui Kikai Co Ltd). Protein precipitation recovered by centrifuge at 1,000*g* for 5 min was suspended in SDS–PAGE sample buffer adjusting to pH 8.8 and then boiled for 95°C, 2 min.

## Southern blotting

Time-course analyses of DNA events in meiosis and cell cycle progression were performed as described previously (Storlazzi et al, 1996; Shinohara et al, 1997). Southern blotting analysis was performed with the same procedure as in Storlazzi et al (1995). For the *HIS4-LEU2* locus, genomic DNA prepared was digested with *Xho*I (for crossover) and *Pst*I (for meiotic DSBs). For the *URA3-ARG4* locus, the DNA was digested with *Xho*I. Probes for Southern blotting were Probe 155 for crossover and Probe 291 for DSB detection on the *HIS4-LEU2* as described in Xu et al (1995). Image Gauge software (Fujifilm Co. Ltd.) was used for quantification for bands.

## PFGE

For PFGE, chromosomal DNA was prepared in agarose plugs as described in Farmer et al (2011) and Bani Ismail et al (2014) and run at 14°C in a CHEF DR-III apparatus (Bio-Rad) using the field 6 V/cm at a 120° angle. Switching times followed a ramp from 15.1 to 25.1 s.

## Statistics

Means ± SD values are shown. Datasets (Figs 2K and 4F) were compared using the Mann–Whitney U-test (Prism, GraphPad).

# Supplementary Information

# Acknowledgements

We thank Drs. Neil Hunter and Andreas Hochwagen for discussion. We are grateful to Drs. Mark Goebl, David Stuart, Michael Lichten, and Doug Koshland for materials used in this study. Z Zhu was supported by scholarship from the graduate school of science in Osaka University. This work was supported by a Grant-in-Aid from the JSPS KAKENHI Grant Number; 22125001, 22125002, 15H05973 and 16H04742, 19H00981 to A Shinohara. M Shinohara was supported by the Japanese Society for the Promotion of Science through the Next Generation World-Leading Researchers program (NEXT).

## Author Contributions

Z Zhu: conceptualization, data curation, formal analysis, investigation, visualization, and writing—review and editing.
M Bani Ismail: data curation, investigation, and writing—review and editing.
M Shinohara: conceptualization, resources, formal analysis, methodology, and writing—review and editing.
A Shinohara: conceptualization, resources, data curation, formal analysis, supervision, funding acquisition, validation, investigation, visualization, project administration, and writing—original draft, review, and editing.

## Conflict of Interest Statement

The authors declare that they have no conflict of interest.

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
