## [Reviewer comments · Life Science Alliance]

Life Science Alliance

SCF(Cdc4) ubiquitin ligase regulates synaptonemal complex formation during meiosis

Zhihui Zhu, Mohammad Bani Ismail, Miki Shinohara, and Akira Shinohara

DOI: <https://doi.org/10.26508/lsa.202000933>

Corresponding author: Akira Shinohara, Institute for Protein Research, Osaka University

Review Timeline:

Submission Date:	2020-10-09
Editorial Decision:	2020-10-12
Revision Received:	2020-11-21
Editorial Decision:	2020-11-23
Revision Received:	2020-11-23
Accepted:	2020-11-24

Scientific Editor: Shachi Bhatt

Transaction Report:

(Note: Please note that the manuscript was previously reviewed at another journal and the reports were taken into account in the decision-making process at Life Science Alliance. With the exception of the correction of typographical or spelling errors that could be a source of ambiguity, letters and reports are not edited. The original formatting of letters and referee reports may not be reflected in this compilation.)

Reviewer #1 Review

Comments to the Authors (Required):

In this manuscript entitled "SCFCdc4 ubiquitin ligase regulates synaptonemal complex formation during meiosis", the authors demonstrated the role of the SCF E3 ubiquitin ligase during meiosis. Using a meiosis-specific depletion of Cdc53 and a temperature-sensitive allele of Cdc4 in budding yeast, the authors showed that SC assembly and the metaphase-to-anaphase-I transition fail in the absence of SCFCdc4. Strikingly, the synapsis defects in *cdc53* and *cdc4* mutants can be suppressed by *pch2* deletion, suggesting that SCF might collaborate with PCH2 to ensure SC assembly.

Meiotic progression in *Cdc53^{mn}* and *Cdc4^{ts}* mutants was analyzed thoroughly, and the evidence presented here supports the main conclusion of this paper. Although the detailed mechanism of SCF's action remains to be determined, this work provides a tantalizing glimpse into the

requirement of SCF in axis/SC assembly and crossover formation. The role of SCF during meiotic prophase has just begun to emerge in mammals (Guan et al., 2020). Thus, it is particularly exciting to see the similar requirement of SCF during meiosis in budding yeast.

Major Points:

My only major concerns are regarding the Discussion. The authors showed that crossover recombination occurs normally at two hotspots in Cdc53mn mutants, while they exhibit severe SC defects. Based on these observations, the authors concluded that "SC, at least SC elongation, IS NOT required for efficient formation of COs per se". However, partial SC stretches still form on some regions of chromosomes in Cdc53mn mutants, and we cannot rule out the possibility that the two hot spots examined are still able to synapse and form crossovers in an SC-dependent manner.

Also, regarding the discussion on the implication for pachytene checkpoint, the proposal that "SC elongation (synapsis) is not monitored by the surveillance mechanism" is not well supported by the current evidence and not likely to be true. Recent work by the Vader lab nicely demonstrates that recombination and synapsis are monitored by a shared checkpoint mechanism in budding yeast and that SC polymerization dictates the mode of Pch2's function (Raina and Vader, 2020). I would suggest the authors take a look at this recent work and incorporate it into the Discussion.

Minor Points:

1. Numerous grammatical and punctuation errors are still present throughout the manuscript. I will list just 3 examples: On page 6, the last sentence ("However, the arrest...") should be connected to the next sentence on page 7 (Since the...) with a comma.

On page 18, in the second sentence of the third paragraph, there is an issue with the plural verb. Please change "there are difference" to "there are differences".

On page 19, in the first sentence of the second paragraph, "defect" should be changed to "detect". In that sentence, "meiosis chromosome spreads" should be changed to "meiotic chromosome spreads".

I suggest the authors go through the text thoroughly and make sure it is free of grammatical errors.

2. On page 7, Cdc5 is described to be expressed "after pachytene exit". However, Cdc5 is expressed during pachytene by Ndt80 and is required for pachytene exit (Clyne et al., 2003; Sourirajan and Lichten, 2008). This needs clarification.

3. On page 7, in the sentence describing Fig 1D ("indicating an arrest at late prophase-1 such as metaphase/anaphase-I transition in the CDC53mn mutant"), prophase 1 is distinct from metaphase/anaphase 1, and it is incorrect to describe the metaphase-to-anaphase transition as part of prophase I.

4. The results shown in Figure S1E don't have labels for fragment sizes. Also, it might be helpful to indicate with arrows what to look for on the gel, especially for readers who are not familiar with CHEF electrophoresis.

5. In Figure 4E, adding labels showing color codes (e.g. red - long lines; green - short lines; blue - dots) would help the readers to understand the graph better.

6. Please label Figure 4F. The authors might want to consider moving Figure S2C next to Figure 4F so that the graph and the actual images can be shown side by side.
7. In Figures 5D, 6C, 6D, and 6F, the purple color for the data points from CDC53mn pch2 double mutants is hard to discern from the blue. The color for the label is more purple, and it is fine.
8. Have the authors examined the localization of axis components and meiotic recombination in cdc4-ts mutant?
9. In Discussion, it was mentioned that the authors couldn't detect Cdc53 by immunofluorescence. If this is unpublished data, it should be noted as "data not shown". Have the authors examined the localization of Cdc4? Given the recent evidence that the mammalian Skp1 is localized along the chromosome axis in mammals, it will be interesting to see whether the budding yeast SCF is also localized to meiotic chromosomes.

Reviewer #2 Review

Comments to the Authors (Required):

Zhu and coworkers examine the role of the SCF E3 ligase in meiotic progression, chromosome morphology/synapsis, and recombination in budding yeast, using a meiotic depletion allele of the cullin Cdc53 (CDC53mn), and a ts allele of the Cdc4 F-box protein. Cells in which Cdc53 is depleted fail to perform meiotic divisions and display synaptonemal complex (SC) defects, but do form crossovers and progress through exit from pachytene. Fewer data are available for cdc4-ts, but these mutant cells have SC defects but progress through meiotic divisions. Interestingly, mutants lacking Pch2, which remodels the meiotic chromosome axis, restore SC formation to CDC53mn cells but impairs DSB repair and abolishes crossover formation, and restore SC formation but reduce meiotic progression in cdc4-ts cells (crossovers were not examined, but the defect most likely is in DSB repair).

The work in this paper is generally well done and documented, and the conclusions are largely supported by the experimental data. It is of interest in that it adds to the growing body of evidence for SCF roles in regulating meiosis. However, the conclusions are limited, as the SCF has a large number of targets, and there are phenotypic differences between CDC53mn, which targets a core SCF component, and cdc4ts, which targets one of the several F-box proteins that confer substrate specificity to the SCF. This suggests that the phenotypes of CDC53mn may reflect loss of activity of several SCF isoforms, not just Cdc4-SCF; alternatively, it is possible that cdc4ts is a hypomorph at 32°C, with some functions impaired more than others. Furthermore, as authors state, CDC53mn appears to impact multiple stages of meiosis (S-phase, break repair, axis and central element morphogenesis, meiotic division), raising the questions of whether these are due to an impact on a single or on multiple SCF targets, and which phenotypes are a direct consequence of loss of SCF activity and which are secondary. Perhaps a proteomic analysis might give some insight, or a cytological investigation asking if Cdc4-SCF is localized to meiotic chromosomes, but of course this is beyond what would be reasonable to request in revision.

A core observation, that pch2 mutants suppress one phenotype (synapsis defects) but exacerbate a second (DSB repair delay/defect), remains perplexing. Are the persistent high levels of DSBs in CDC53mn due to DSBs that form early but persist unrepaired, or are they due to continued DSB

formation in the absence of axis remodeling? Further analysis (of noncrossover formation, joint molecule formation, of DSB kinetics in *sae2* or *dmc1* mutants) might provide direction toward an answer, but again requesting this much additional work would seem not to be reasonable.

The work reported in this paper will be of interest to the meiosis community, in that it provides evidence for multiple processes regulated by SCF complexes that will be ground for future mechanistic investigation. Its main value lies in the careful reporting of mutant phenotypes.

Minor comments:

1. While DNA studies appear to have been done three times, the absence of error bars in time-course cytological studies suggest that these experiments were only done once; the same question arises with regards to flow cytometry measures of DNA content.
2. In a related matter, in Figure 1H and Figure 2J, Rad51 foci are delayed in CDC53mn relative to wild type. But in Figure 6D they appear at the same time, although they do persist longer in the mutant. It would be worth knowing how much experiment-to-experiment variation there was in timing, at least for strains that were examined more than once.
3. A legend would be helpful for Figure 4E.
4. The pulsed field gels in Supplementary Figure 1E are not very clear, and this method is not so great for monitoring DSB formation and repair, anyway. Given that CDC53mn cells at 4h are still in S phase, this smeariness may reflect chromosome replication rather than DSB formation; certainly the conclusion that reappearance of bands at 5h reflects repair is dissonant with the kinetics of Rad51 foci, which peak at 5-6h in this strain, and with southern blots, which show that DSBs also peak at 5-6h.

Reviewer #3 Review

Comments to the Authors (Required):

The manuscript "SCF(Cdc4) ubiquitin ligase regulates synaptonemal complex formation during meiosis" by Zhu et al. presents a series of genetic experiments in yeast probing the role of the Cdc4-dependent SCF ubiquitin ligase complex in meiotic prophase of the budding yeast, *S. cerevisiae*. As this complex is required for cell growth, the authors take advantage of a "meiotic null" (MN) allele of the Cdc53 subunit that they created. They show that Cdc53-MN induces a delay in the timing of pre-meiotic DNA replication (S phase), and concomitant delays in the steps of meiotic recombination (double-strand break formation, crossover formation). While meiotic recombination occurs at near-wild-type levels, inter-homolog synapsis is defective, as evidenced by shorter synaptonemal complex (SC) assemblies on chromosomes and the presence of poly-complexes, large aggregates of SC proteins that form when their chromosome loading is defective. The authors also show that deletion of Pch2, a AAA+ ATPase regulator of the Hop1 meiotic axis protein, suppresses the defect in synaptonemal complex assembly. Last, they also introduced a mutation in CDC4, the F-box protein in SCF, and showed similar SC assembly defects like CDC53, suggesting they work together under SCF for regulating SC formation. However, unlike CDC53-MN, CDC4 mutants can go through meiosis I, suggesting they play different roles in this regulation.

This paper deals with an interesting topic, and is conceptually related to two recent manuscripts that showed a role for a homologous complex in mouse meiosis (Gray 2020 and Guan 2020, both cited in the manuscript). The experiments seem to be well performed. But I have two major concerns that would be critical for publication in this journal. First and foremost, there is little in the way of mechanistic insight from the presented experiments: the phenotypes shown are mostly intermediate (delays in various steps, e.g.) and the molecular function of the SCF ubiquitin ligase

complex (i.e. its substrate in meiosis) is not defined. Second, and related to the above, the writing and the presentation of the experiments is very difficult to follow, even for an expert in the cell cycle and meiotic prophase. Below I note several specific concerns that the authors may want to consider for preparing a revision, for this or another journal.

Major concerns:

Based on previous studies (Qiao et al., 2014), (Gray et al., 2020), and (Guan et al., 2020), it seems suggesting that the SCF has conserved role in recombination regulation and meiosis transition regulation, as pointed out in this manuscript. However, it is not clear to me in this manuscript what specific role the SCF ubiquitin ligase plays to regulate other protein factors during meiosis prophase in yeast. It seems that SCF delays meiosis onset through Sic1, does the inducible Sic1 knockout have similar SC assembly defects? I am wondering if the delayed meiosis onset caused a defect in the SC assembly, as the axis proteins like Hop1 are still expressed in a similar pattern to WT.

The authors showed that the CDC53mn mutant uncouples crossover formation from SC assembly, which is a novel finding. But it lacks direct evidence why the SC assembly is defective under CDC53mn, and how the Msh4 and Msh5 are properly recruited onto chromosome without other ZMM proteins. The authors have a good discussion speculating on the possible working models, but it would be good to be narrowed and focused on testing out these models.

Regarding the functional linkage with PCH2, the authors may want to cite and discuss a recent paper by Raina and Vader (Current Biology S0960-9822(20)31254-9). This work lays out a feedback loop in which chromosome-localized Hop1 promotes Zip1 (SC) assembly, which in turn recruits Pch2 to down-regulate Hop1's chromosome localization. Because recombination and SC assembly depend so heavily on Hop1's recruitment to chromosomes, it seems possible (likely?) that the pch2 mutant suppresses the phenotypes of CDC53mn by increasing Hop1 on chromosomes. In that case, these mutants may not be as functionally linked as the authors suggest. Indeed, the authors show that CDC53mn does not affect Pch2 levels - so why is Pch2 diagrammed in the authors' models as being regulated by SCF?

Some minor points:

Figure S1D: the loading amount is not well controlled so that it's hard to conclude Cdc6 can retain up to 12 hours in CDC53mn after meiosis.

Figure 1D: It looks like the Cdc53-MN mutant is starting to undergo meiotic divisions by the 10-12 hour time points, but then the analysis ends at 12 hours. Would these cells go through meiosis if given more time? In other words, is this simply a long delay in progression rather than an arrest?

Figure 1E: we can see clear difference between panel ii and iii for the WT, suggesting normal meiosis progression, however, I don't see difference between ii and v in terms of zip1 line length except for the "poly-complex". It would be good to show another panel for the mutant that progress further into meiosis.

Figure 1F: again, how do you define the Zip1 line length as long or short?

Figure 4F: Not labeled in the panel. Also, based on the 2 hour delay of meiosis/DSB formation, the comparison should be 4h of WT and 6h of mutant, rather than 5h of mutant for Figure 4F.

Figure 5A: pch2 mutant and pch2 cdc53mn have similar phenotype, suggesting CDC53mn has no extra effect on SC assembly? Or Cdc53 involves in the downstream pathway, if Pch2 and Cdc53 are in the same regulatory pathway?

Time points throughout the manuscript varies for comparison of WT and mutants, for example, Figure S4.D used 3 hour difference for WT and mutants, while in Figure 4F used 1 hour difference for WT and mutant, and some other figures showing 2 hour difference. It's hard to compare fairly the phenotypes between the WT and mutants. In this regard, it's good to include labels for the meiosis prophase stages in the immuno-staining images throughout the paper to make it easier for readers to follow.

October 12, 2020

Re: Life Science Alliance manuscript #LSA-2020-00933-T

Prof. Akira Shinohara
Institute for Protein Research, Osaka University
Dept. of Integrated Protein Functions
3-2 Yamadaoka
Suita, Osaka 565-0871
Japan

Dear Dr. Shinohara,

Thank you for transferring your manuscript entitled "SCF(Cdc4) ubiquitin ligase regulates synaptonemal complex formation during meiosis" to Life Science Alliance (LSA).

For a brief overview, the manuscript was peer-reviewed by a panel of 3 experts at a LSA partner journal, where the reviewers found the findings to be interesting and novel, but were concerned about lack of sufficient mechanistic depth, despite two round of review. The authors transferred the manuscript and reviewers comments to LSA with the help of the editors from the partner journal. At LSA, the conceptual advance and the findings showing the regulation of synaptonemal complex formation during meiosis were considered sufficient for publication, pending minor revisions.

For publication in LSA, you should

- address the discussion concerns raised by Rev 1
- discuss the alternate possibilities for the observed phenotypes as pointed out by Rev 2 (paragraph 2)
- Further analysis to figure out the mechanistic details, i.e. decipher the opposing effects of Pch2 mutant on synapsis phenotype and DSB repair phenotype (Rev 2 paragraph 3); how CDC53mn mutant uncouples crossover function from SC assembly (Rev 3 pt 1) will not be necessary for publication in LSA.
- Discuss the Current Biology paper as suggested by Rev 3 (pt 2)
- address all the minor points raised by all 3 reviewers, except Rev 3's minor comments 2, 3, and 6 (Fig 1D, 1E and 5A) - which should only be addressed with new data if the data is readily available, otherwise a discussion should be enough
- edit the manuscript for grammatical and sentence structure errors
- submit a point-by-point rebuttal to all the reviewers' concerns

While you are revising your manuscript, please also attend to the below editorial points to help

expedite the publication of your manuscript. Please direct any editorial questions to the journal office.

Thank you for this interesting contribution to Life Science Alliance. We are looking forward to receiving your revised manuscript.

Sincerely,

Shachi Bhatt, Ph.D.
Executive Editor
Life Science Alliance
<https://www.lsa-journal.org/>
Tweet @SciBhatt @LSAJournal

B. MANUSCRIPT ORGANIZATION AND FORMATTING:

Life Science Alliance manuscript #LSA-2020-00933-T

Dear Dr. Shachi Bhatt,

Thanks for reviewing our paper and giving us constructive comments. We have been working hard to revise our paper and now are able to respond to all of the comments, as described below.

We hope that this revised version of our paper is suitable for publication in *Life Science Alliances*.

Our responses to reviewers/editors are shown in green below.

Thanks for your help on our paper.

Sincerely yours,

Akira Shinohara

Responses to reviewers:

Comments by the editor;

- address the discussion concerns raised by Rev 1

>We addressed this. See below.

- discuss the alternate possibilities for the observed phenotypes as pointed out by Rev 2 (paragraph 2)

>We discussed the alternative possibility. See below.

- Further analysis to figure out the mechanistic details, i.e. decipher the opposing effects of Pch2 mutant on synapsis phenotype and DSB repair phenotype (Rev 2 paragraph 3); how CDC53mn mutant uncouples crossover function from SC assembly (Rev 3 pt 1) will not be necessary for publication in LSA.

>As suggested, we do not address these issues, but added comments as shown below..

- Discuss the Current Biology paper as suggested by Rev 3 (pt 2)

>We discussed the alternative possibility. See below.

- address all the minor points raised by all 3 reviewers, except Rev 3's minor comments 2, 3, and 6 (Fig 1D, 1E and 5A) - which should only be addressed with new data if the data is readily available, otherwise a discussion should be enough

>We addressed all minor points. See below.

- edit the manuscript for grammatical and sentence structure errors

>We asked a professional English Editing service named "Egitage" to correct our English. The proof from the company is attached in the end of this file.

- submit a point-by-point rebuttal to all the reviewers' concerns

>Please check below.

Reviewer #1 (Comments to the Authors (Required)):

In this manuscript entitled "SCFCdc4 ubiquitin ligase regulates synaptonemal complex formation during meiosis", the authors demonstrated the role of the SCF E3 ubiquitin ligase during meiosis. Using a meiosis-specific depletion of Cdc53 and a temperature-sensitive allele of Cdc4 in budding yeast, the authors showed that SC assembly and the metaphase-to-anaphase-I transition fail in the absence of SCFCdc4. Strikingly, the synapsis defects in *cdc53* and *cdc4* mutants can be suppressed by *pch2* deletion, suggesting that SCF might collaborate with PCH2 to ensure SC assembly.

Meiotic progression in *Cdc53^{mn}* and *Cdc4^{-ts}* mutants was analyzed thoroughly, and the evidence presented here supports the main conclusion of this paper. Although the detailed mechanism of SCF's action remains to be determined, this work provides a tantalizing glimpse into the requirement of SCF in axis/SC assembly and crossover formation. The role of SCF during meiotic prophase has just begun to emerge in mammals (Guan et al., 2020). Thus, it is particularly exciting to see the similar requirement of SCF during meiosis in budding yeast.

Major Points:

My only major concerns are regarding the Discussion. The authors showed that crossover recombination occurs normally at two hotspots in *Cdc53^{mn}* mutants, while they exhibit severe SC defects. Based on these observations, the authors concluded that "SC, at least SC elongation, IS NOT required for efficient formation of COs per se". However, partial SC stretches still form on some regions of chromosomes in *Cdc53^{mn}* mutants, and we cannot rule out the possibility that the two hot spots examined are still able to synapse and form crossovers in an SC-dependent manner.

-We rewrote a part of Discussion by pointing out the possibility raised by this reviewer as an alternative possibility and also turning down our preferred idea (page 22, line 1-5).

Also, regarding the discussion on the implication for pachytene checkpoint, the proposal that "SC elongation (synapsis) is not monitored by the surveillance mechanism" is not well supported by the current evidence and not likely to be true. Recent work by the Vader lab nicely demonstrates that recombination and synapsis are monitored by a shared checkpoint mechanism in budding yeast and that SC polymerization dictates the mode of Pch2's function (Raina and Vader, 2020). I would suggest the authors take a look at this recent work and incorporate it into the Discussion.

-Since, as pointed by the reviewer, our original proposal that SC elongation is not monitored by the surveillance mechanism is not "fully" supported by the experiments and is not consistent with the idea shown by the recent paper (Raina and Vader 2020), we deleted our original proposal from the main text. In addition, we incorporated the observation by Raina and Vader to explain how

the *PCH2* deletion suppresses SC defect in the *CDC53mn* mutant (page 20, second paragraph, last 6 lines).

Minor Points:

1. Numerous grammatical and punctuation errors are still present throughout the manuscript. I will list just 3 examples: On page 6, the last sentence ("However, the arrest...") should be connected to the next sentence on page 7 (Since the...) with a comma.

-We corrected them and asked an English editing service (Edigate Co. Ltd.) to correct the grammatical errors in the full text.

On page 18, in the second sentence of the third paragraph, there is an issue with the plural verb. Please change "there are difference" to "there are differences".

-We corrected it.

On page 19, in the first sentence of the second paragraph, "defect" should be changed to "detect". In that sentence, "meiosis chromosome spreads" should be changed to "meiotic chromosome spreads".

-We corrected them.

I suggest the authors go through the text thoroughly and make sure it is free of grammatical errors.

-To avoid grammatical errors, as mentioned, we asked English editing service to correct the errors in full text.

2. On page 7, *Cdc5* is described to be expressed "after pachytene exit". However, *Cdc5* is expressed during pachytene by *Ndt80* and is required for pachytene exit (Clyne et al., 2003; Sourirajan and Lichten, 2008). This needs clarification.

-We replaced pachytene exit with "mid-pachytene stage".

3. On page 7, in the sentence describing Fig 1D ("indicating an arrest at late prophase-1 such as metaphase/anaphase-I transition in the *CDC53mn* mutant"), prophase 1 is distinct from metaphase/anaphase 1, and it is incorrect to describe the metaphase-to-anaphase transition as part of prophase I.

-We rephrased the sentence by deleting late prophase I here.

4. The results shown in Figure S1E don't have labels for fragment sizes. Also, it might be helpful to indicate with arrows what to look for on the gel, especially for readers who are not familiar with CHEF electrophoresis.

-We added some labels in Figure S1E for clarification and removed words in the legends.

5. In Figure 4E, adding labels showing color codes (e.g. red - long lines; green - short lines; blue - dots) would help the readers to understand the graph better.

-We added color codes like in Figure 1F.

6. Please label Figure 4F. The authors might want to consider moving Figure S2C next to Figure 4F so that the graph and the actual images can be shown side by side.

-We added label of Figure 4G (since we added images as Figure 4F) and new images of GFP/DAPI in Figure 4F.

7. In Figures 5D, 6C, 6D, and 6F, the purple color for the data points from CDC53mn pch2 double mutants is hard to discern from the blue. The color for the label is more purple, and it is fine.

-We changed color code with more purple.

8. Have the authors examined the localization of axis components and meiotic recombination in cdc4-ts mutant?

-No, we have not analyzed it. We do not think this adds more insight than what is described in the paper.

9. In Discussion, it was mentioned that the authors couldn't detect Cdc53 by immunofluorescence. If this is unpublished data, it should be noted as "data not shown". Have the authors examined the localization of Cdc4? Given the recent evidence that the mammalian Skp1 is localized along the chromosome axis in mammals, it will be interesting to see whether the budding yeast SCF is also localized to meiotic chromosomes.

-We tried to check the localization of Cdc53 by anti-Cdc53 and Cdc53-Flag by anti-Flag, but could not detect the localization. This is written in Page 19, second paragraph, line 1-2.

Reviewer #2 (Comments to the Authors (Required)):

Zhu and coworkers examine the role of the SCF E3 ligase in meiotic progression, chromosome morphology/synapsis, and recombination in budding yeast, using a meiotic depletion allele of the cullin Cdc53 (CDC53mn), and a ts allele of the Cdc4 F-box protein. Cells in which Cdc53 is depleted fail to perform meiotic divisions and display synaptonemal complex (SC) defects, but do form crossovers and progress through exit from pachytene. Fewer data are available for cdc4-ts, but these mutant cells have SC defects but progress through meiotic divisions. Interestingly, mutants lacking Pch2, which remodels the meiotic chromosome axis, restore SC formation to CDC53mn cells but impairs DSB repair and abolishes crossover formation, and restore SC formation but reduce meiotic progression in cdc4-ts cells (crossovers were not examined, but the defect most likely is in DSB repair).

The work in this paper is generally well done and documented, and the conclusions are largely supported by the experimental data. It is of interest in that it adds to the growing body of evidence for SCF roles in regulating meiosis. However, the conclusions are limited, as the SCF has a large number of targets, and there are phenotypic differences between CDC53mn, which targets a core

SCF component, and *cdc4ts*, which targets one of the several F-box proteins that confer substrate specificity to the SCF. This suggests that the phenotypes of *CDC53mn* may reflect loss of activity of several SCF isoforms, not just Cdc4-SCF; alternatively, it is possible that *cdc4ts* is a hypomorph at 32°C, with some functions impaired more than others. Furthermore, as authors state, *CDC53mn* appears to impact multiple stages of meiosis (S-phase, break repair, axis and central element morphogenesis, meiotic division), raising the questions of whether these are due to an impact on a single or on multiple SCF targets, and which phenotypes are a direct consequence of loss of SCF activity and which are secondary. Perhaps a proteomic analysis might give some insight, or a cytological investigation asking if Cdc4-SCF is localized to meiotic chromosomes, but of course this is beyond what would be reasonable to request in revision.

A core observation, that *pch2* mutants suppress one phenotype (synapsis defects) but exacerbate a second (DSB repair delay/defect), remains perplexing. Are the persistent high levels of DSBs in *CDC53mn* due to DSBs that form early but persist unrepaired, or are they due to continued DSB formation in the absence of axis remodeling? Further analysis (of noncrossover formation, joint molecule formation, of DSB kinetics in *sae2* or *dmc1* mutants) might provide direction toward an answer, but again requesting this much additional work would seem not to be reasonable.

-(The editor does not request to answer this). We agree with the reviewer's comments. The phenotypes of the *pch2 CDC53mn* mutants are perplexing, but interesting. Further characterization would be interesting and critical to understand the phenotypes. However, as pointed out, this is out of scope of the paper and should be addressed in future along with the identification of a target(s) of SCF ligase complexes during meiosis.

The work reported in this paper will be of interest to the meiosis community, in that it provides evidence for multiple processes regulated by SCF complexes that will be ground for future mechanistic investigation. Its main value lies in the careful reporting of mutant phenotypes.

Minor comments:

1. While DNA studies appear to have been done three times, the absence of error bars in time-course cytological studies suggest that these experiments were only done once; the same question arises with regards to flow cytometry measures of DNA content.

-We added the number of each experiment in Figure legends. Even for cytology, if it is important, we added error bars in Figure 4E. We would like to stress quantification of Zip1 staining in the *CDC53mn* mutant in Figure 1F is shown in other experiments in Figure 3H and 5C (and Figure S3C) with different co-staining proteins

2. In a related matter, in Figure 1H and Figure 2J, Rad51 foci are delayed in *CDC53mn* relative to wild type. But in Figure 6D they appear at the same time,

although they do persist longer in the mutant. It would be worth knowing how much experiment-to-experiment variation there was in timing, at least for strains that were examined more than once.

-We added the number of each experiment in Figure legends.

3. A legend would be helpful for Figure 4E.

-We added more words in legend of Figure 4E and labels in Figure 4E by pointed out by #1 reviewer.

4. The pulsed field gels in Supplementary Figure 1E are not very clear, and this method is not so great for monitoring DSB formation and repair, anyway. Given that CDC53mn cells at 4h are still in S phase, this smeariness may reflect chromosome replication rather than DSB formation; certainly the conclusion that reappearance of bands at 5h reflects repair is disconsonant with the kinetics of Rad51 foci, which peak at 5-6h in this strain, and with southern blots, which show that DSBs also peak at 5-6h.

-This CHEF analysis was requested by the reviewer in first round submission to the other journal since we looked DSB repair in the *pch2 CDC53mn* mutant cells only at *HIS4-LEU2* locus. We agree the results are not convincing for mutants such as the *pch2* and *CDC53mn* single mutants, but is clear for the *pch2 CDC53mn* double mutant, which showed little recovery of full-length sizes of chromosomes at late times. So we do not think that Southern blotting analysis of DSB repair is necessary and will add new information.

Reviewer #3 (Comments to the Authors (Required)):

The manuscript "SCF(Cdc4) ubiquitin ligase regulates synaptonemal complex formation during meiosis" by Zhu et al. presents a series of genetic experiments in yeast probing the role of the Cdc4-dependent SCF ubiquitin ligase complex in meiotic prophase of the budding yeast, *S. cerevisiae*. As this complex is required for cell growth, the authors take advantage of a "meiotic null" (MN) allele of the Cdc53 subunit that they created. They show that Cdc53-MN induces a delay in the timing of pre-meiotic DNA replication (S phase), and concomitant delays in the steps of meiotic recombination (double-strand break formation, crossover formation). While meiotic recombination occurs at near-wild-type levels, inter-homolog synapsis is defective, as evidenced by shorter synaptonemal complex (SC) assemblies on chromosomes and the presence of poly-complexes, large aggregates of SC proteins that form when their chromosome loading is defective. The authors also show that deletion of Pch2, a AAA+ ATPase regulator of the Hop1 meiotic axis protein, suppresses the defect in synaptonemal complex assembly. Last, they also introduced a mutation in CDC4, the F-box protein in SCF, and showed similar SC assembly defects like CDC53, suggesting they work together under SCF for regulating SC formation. However, unlike CDC53-MN, CDC4 mutants can go through meiosis I, suggesting they play different roles in this regulation.

This paper deals with an interesting topic, and is conceptually related to two recent manuscripts that showed a role for a homologous complex in mouse

meiosis (Gray 2020 and Guan 2020, both cited in the manuscript). The experiments seem to be well performed. But I have two major concerns that would be critical for publication in this journal. First and foremost, there is little in the way of mechanistic insight from the presented experiments: the phenotypes shown are mostly intermediate (delays in various steps, e.g.) and the molecular function of the SCF ubiquitin ligase complex (i.e. its substrate in meiosis) is not defined. Second, and related to the above, the writing and the presentation of the experiments is very difficult to follow, even for an expert in the cell cycle and meiotic prophase. Below I note several specific concerns that the authors may want to consider for preparing a revision, for this or another journal.

Major concerns:

Based on previous studies (Qiao et al., 2014), (Gray et al., 2020), and (Guan et al., 2020), it seems suggesting that the SCF has conserved role in recombination regulation and meiosis transition regulation, as pointed out in this manuscript. However, it is not clear to me in this manuscript what specific role the SCF ubiquitin ligase plays to regulate other protein factors during meiosis prophase in yeast. It seems that SCF delays meiosis onset through Sic1, does the inducible Sic1 knockout have similar SC assembly defects? I am wondering if the delayed meiosis onset caused a defect in the SC assembly, as the axis proteins like Hop1 are still expressed in a similar pattern to WT.

-We do not think that Sic1 is responsible for SC formation and meiotic recombination, since the *CDC53mn* mutant did degrade Sic1 (shown in Figure S1D and S4B) to undetectable levels with substantial delay. To address a possible negative role of Sic1 during meiosis (not positive role proposed by this reviewer), over-expression (OE) of Sic1 during meiosis is one way to analyze. A previous study by Sawarynski et al., (PNAS 2009 106 (1) 232237) showed that the OE of Sic1 induces DNA re-replication during meiosis, which we did not see in the *CDC53mn* mutant (Figure S1B and S4A) even with stabilization of Cdc6, DNA replication initiator protein in the *CDC53mn* mutant. Moreover, we could not deny a possibility that an early defect in the *CDC53mn* mutant (e.g. during pre-meiotic S phase) may cause various defect during prophase I. On the other hand, we do see normal meiotic recombination in the *CDC53mn* mutant, which, like SC assembly, may be also affected by the early meiotic event I. If it is likely, we need more complicated explanation on how *PCH2* deletion suppresses SC defects in the *CDC53mn* mutant, since there is little report on an early role of Pch2.

The authors showed that the *CDC53mn* mutant uncouples crossover formation from SC assembly, which is a novel finding. But it lacks direct evidence why the SC assembly is defective under *CDC53mn*, and how the Msh4 and Msh5 are properly recruited onto chromosome without other ZMM proteins. The authors have a good discussion speculating on the possible working models, but it would be good to be narrowed and focused on testing out these models.

-(The editor does not request to answer this). Although this reviewer suggests the recruitment of Msh4/5 foci without other ZMMs, we think that is unlikely. As shown in Figure 2I and J and Figure S2A, we could not detect ZMM foci except

Msh4/5 foci in the *CDC53mn* mutant, we could interpret this by assuming that the mutant might assemble functional (invisible) ZMM on meiotic chromosomes since the mutant is proficient in ZMM-dependent CO formation; e.g. CO formation in the *CDC53mn* mutant still depends on Zip3 (Figure 3A-F). To avoid this kind of misunderstanding, we re-wrote related parts in Discussion (page 22, first paragraph line 5-12).

Regarding the functional linkage with PCH2, the authors may want to cite and discuss a recent paper by Raina and Vader (Current Biology S0960-9822(20)31254-9). This work lays out a feedback loop in which chromosome-localized Hop1 promotes Zip1 (SC) assembly, which in turn recruits Pch2 to down-regulate Hop1's chromosome localization. Because recombination and SC assembly depend so heavily on Hop1's recruitment to chromosomes, it seems possible (likely?) that the *pch2* mutant suppresses the phenotypes of *CDC53mn* by increasing Hop1 on chromosomes. In that case, these mutants may not be as functionally linked as the authors suggest. Indeed, the authors show that *CDC53mn* does not affect Pch2 levels - so why is Pch2 diagrammed in the authors' models as being regulated by SCF?

-We rewrote the Discussion by suggesting an alternative possibility to explain how the *PCH2* deletion suppresses SC defects in the *CDC53mn* mutant based on the recent Current Biology paper (Raina and Vader 2020). Simply more loading of Hop1 on chromosomes by the *pch2* deletion could indirectly promote SC formation in the *CDC53mn* and *cdc4-ts* mutants (page 20, second paragraph, last 6 lines). We also town down our model (Cdc53's target X works together with Pch2 in Figure 7) in the new version.

Some minor points:

Figure S1D: the loading amount is not well controlled so that it's hard to conclude Cdc6 can retain up to 12 hours in the *CDC53mn* after meiosis.

-It is very clear that Cdc6 accumulates in the *CDC53mn* mutant in multiple experiments (shown in the *CDC53mn* and the *CDC53mn* mutants in Figure S1B an 4A).

Figure 1D: It looks like the Cdc53-MN mutant is starting to undergo meiotic divisions by the 10-12 hour time points, but then the analysis ends at 12 hours. Would these cells go through meiosis if given more time? In other words, is this simply a long delay in progression rather than an arrest?

-(The editor does not request to answer this). We analyzed meiotic progression by 24 h by DAPI staining and found that the mutant shows one nucleus in a cell at 24 h (shown in Fig. 3A and 6A), indicating very tight arrest of meiosis progression.

Figure 1E: we can see clear difference between panel ii and iii for the WT, suggesting normal meiosis progression, however, I don't see difference between ii and v in terms of zip1 line length except for the "poly-complex". It would be good to show another panel for the mutant that progress further into meiosis.

-The editor does not request to answer this). We classified ii and v in Figure 1E as a partially elongated Zip1 staining. However, v shows Zip1-poly complex as pointed. As mentioned, in the text, after this stage of “v”, these Zip1-containing structures disappeared together with disappearance of Rad51 foci at late stages; e.g. Zip1-negative/Rad51 negative cells are in next stage as shown in the kinetical analysis of Zip1/Rad51 in Figure 1F-H.

Figure 1F: again, how do you define the Zip1 line length as long or short?

-We defined long and short as spreads containing more than 10 Zip1 lines and less than 10 lines, respectively. This criterion is written in the legend of Figure 1E.

Figure 4F: Not labeled in the panel. Also, based on the 2 hour delay of meiosis/DSB formation, the comparison should be 4h of WT and 6h of mutant, rather than 5h of mutant for Figure 4F.

-We added a label of Figure 4“F” as pointed. The previous description of “5 h” in the mutant in the main text was wrong. As shown in legend of Figure 4F, “6 h” is a right time point. Thus, main text was corrected from 5 h to 6 h.

Figure 5A: *pch2* mutant and *pch2 cdc53mn* have similar phenotype, suggesting *CDC53mn* has no extra effect on SC assembly? Or *Cdc53* involves in the downstream pathway, if *Pch2* and *Cdc53* are in the same regulatory pathway?

-(The editor does not request to answer this). At least from Zip1 staining (and Hop1 staining), SC formed in the *CDC53mn* and the *pch2 CDC53mn* mutant are very similar. Thus, it is hard to conclude any differences at these levels. As pointed by Raina and Vader (2020), we added a possibility that *Pch2* functions upstream of *Cdc53* by regulating Hop1 protein (see above). For recombination/DSB repair, *Pch2* and *Cdc53* function in distinct pathways.

Time points throughout the manuscript varies for comparison of WT and mutants, for example, Figure S4.D used 3 hour difference for WT and mutants, while in Figure 4F used 1 hour difference for WT and mutant, and some other figures showing 2 hour difference. It's hard to compare fairly the phenotypes between the WT and mutants. In this regard, it's good to include labels for the meiosis prophase stages in the immuno-staining images throughout the paper to make it easier for readers to follow.

-We added labels of leptonema, zygonema and pachynema in Figures.

November 23, 2020

RE: Life Science Alliance Manuscript #LSA-2020-00933-TR

Prof. Akira Shinohara
Institute for Protein Research, Osaka University
Dept. of Integrated Protein Functions
3-2 Yamadaoka
Suita, Osaka 565-0871
Japan

Dear Dr. Shinohara,

Thank you for submitting your revised manuscript entitled "SCF(Cdc4) ubiquitin ligase regulates synaptonemal complex formation during meiosis". We would be happy to publish your paper in Life Science Alliance pending final revisions necessary to meet our formatting guidelines.

Along with the points listed below, please also attend to the following:

- please upload your supplementary figures as single files and upload your tables as editable doc or excel files
- please use the [10 author names, et al.] format in your references (i.e. limit the author names to the first 10)
- would it be possible to label each cell cycle stage within the figure 1E, instead of labeling each row as i,ii,iii..?

A. FINAL FILES:

B. MANUSCRIPT ORGANIZATION AND FORMATTING:

Sincerely,

Shachi Bhatt, Ph.D.
Executive Editor
Life Science Alliance
<https://www.lsjournal.org/>
Tweet @SciBhatt @LSAJournal

November 24, 2020

RE: Life Science Alliance Manuscript #LSA-2020-00933-TRR

Prof. Akira Shinohara
Institute for Protein Research, Osaka University
Dept. of Integrated Protein Functions
3-2 Yamadaoka
Suita, Osaka 565-0871
Japan

Dear Dr. Shinohara,

Thank you for submitting your Research Article entitled "SCF(Cdc4) ubiquitin ligase regulates synaptonemal complex formation during meiosis". It is a pleasure to let you know that your manuscript is now accepted for publication in Life Science Alliance. Congratulations on this interesting work.

DISTRIBUTION OF MATERIALS:

Again, congratulations on a very nice paper. I hope you found the review process to be constructive and are pleased with how the manuscript was handled editorially. We look forward to future exciting submissions from your lab.

Sincerely,

Shachi Bhatt, Ph.D.

Executive Editor

Life Science Alliance

<https://www.lsjournal.org/>
